# Towards enhanced functionality of vagus neuroprostheses through in silico optimized stimulation

Federico Ciotti [1,8], Robert John [1,8], Natalija Katic Secerovic [1,2,8], Noemi Gozzi [1], Andrea Cimolato[1], Naveen Jayaprakash[3,4], Weiguo Song[3,4], Viktor Toth[3,4], Theodoros Zanos [3,4,5,6], Stavros Zanos [3,4,5,6] & Stanisa Raspopovic [1,7]

Bioelectronic therapies modulating the vagus nerve are promising for cardiovascular, inflammatory, and mental disorders. Clinical applications are however limited by side-effects such as breathing obstruction and headache caused by non-specific stimulation. To design selective and functional stimulation, we engineered VaStim, a realistic and efficient *in-silico* model. We developed a protocol to personalize VaStim in-vivo using simple muscle responses, successfully reproducing experimental observations, by combining models with trials conducted on five pigs. Through optimized algorithms, VaStim simulated the complete fiber population in minutes, including often omitted unmyelinated fibers which constitute 80% of the nerve. The model suggested that all Aα-fibers across the nerve affect laryngeal muscle, while heart rate changes were caused by B-efferents in specific fascicles. It predicted that tripolar paradigms could reduce laryngeal activity by 70% compared to typically used protocols. VaStim may serve as a model for developing neuromodulation therapies by maximizing efficacy and specificity, reducing animal experimentation.

The vagus nerve (VN) propagates from cervical to abdominal body regions and, as part of the autonomous nervous system, plays a critical role in regulating the function of various organs[1]. It spreads from more rostral areas, innervating heart, lungs, and laryngeal muscles, toward caudal areas, controlling stomach, pancreas, and intestine[2,3]. The cervical VN is a surgically accessible target[4] and its stimulation is clinically practiced as an alternative to pharmaceuticals for drug-resistant epilepsy, depression, and obstructive sleep apnea[5,6]. While these remarkable results are fueling the translation of this neuromodulation approach in several clinical fields[7], like the cardiovascular[8–10] and gastrointestinal[11], the currently used non-precise stimulation induces multiple side effects, such as cough, neck muscle contractions, dyspnea, swallowing difficulties, headache or throat pain[8,9,12,13]. This significantly limits the safety and efficiency of different VNS approaches[14] and restricts exploration of further therapeutic effects, which slows down the possible clinical translation for human use.

The cervical VN contains thousands of sensory (afferent) and motor (efferent) myelinated and unmyelinated fibers organized into fascicles, with different fiber diameters and specific physiological function[15]. Motor and sensory fibers occupy separate fascicles at higher cervical levels that gradually merge toward the thoracic region. In contrast, the organ-specific fascicles innervating laryngeal muscle,

[1]Laboratory for Neuroengineering, Department of Health Sciences and Technology, Institute for Robotics and Intelligent Systems, ETH Zürich, Zürich, Switzerland. [2]The Mihajlo Pupin Institute, University of Belgrade, Belgrade, Serbia. [3]Northwell Health, New Hyde Park, NY, USA. [4]Feinstein Institutes for Medical Research, Manhasset, NY, USA. [5]Donald and Barbara Zucker School of Medicine at Hofstra/Northwell, Hempstead, NY, USA. [6]Elmezzi Graduate School of Molecular Medicine, Manhasset, NY, USA. [7]Center for Medical Physics and Biomedical Engineering, Medical University of Vienna, Vienna, Austria. [8]These authors contributed equally: Federico Ciotti, Robert John, Natalija Katic Secerovic. ✉e-mail: nesta.fale@gmail.com

lungs, and heart, become distinct toward the thoracic region[16]. Therefore, the controlled modulation of VN activity at the middle cervical level can potentially activate either efferent or afferent pathways and modulate multiple organ-specific outcomes. This makes it a promising neuromodulation target.

Due to the complex structure of the VN and limited a priori knowledge, it is extremely challenging to achieve the selective modulation of an intended physiological function. Indeed, stimulation policies are currently naïve regarding the nerve's composition, such as fascicular anatomy and type-specific fiber diameters, alongside their resulting function. Clinically used stimulation policies thus far do not exhaustively leverage the complex potential fields that could arise from multipolar and multimodal electrical stimulation (combining different parameters of electrical stimulation such as varying frequencies, amplitudes, or waveforms)[17]. Considering the sheer multi-dimensionality of the stimulation parameters space, exploring complex VNS paradigms in-vivo is nearly impossible and thus constrained to explore just a sub-optimal subset of possible policies, such as only bipolar[18]. There is an unmet need to identify alternative ways to test multiple VNS approaches for guiding the design of novel paradigms toward the most effective and selective ones, accelerating the clinical translation of novel VNS applications. In this context, the development of computational models of the VN is particularly promising[19–22]. Previous studies have leveraged computational models to design more spatially selective electrodes and stimulation policies. However, due to the lack of accurate information regarding fiber distributions and function, animal experiments were necessary to assess the physiological effects of novel stimulation paradigms[23,24].

However, currently available models suffer from several limitations. First, they represent nerve fascicles with linearly extruded tubes[21,24–27]. These models are highly unrealistic, as previous research show that fascicles are merging and branching to a large degree along the rostro-caudal axis[16,28,29]. This introduces errors in estimated voltage distribution caused by the stimulation and resulting nerve activity[30]. Moreover, existing models do not account for the realistic distribution and clustering of different fiber types[16]. Consequently, they cannot estimate correctly the spatial fiber activation of specific fiber types, which is crucial as different fiber types modulate specific functions, and as the VN shows organ-specific fascicular organization in big animals[16,28,31]. Furthermore, although the majority of VN fibers are unmyelinated ones (approx. 60–80%)[3], they are mainly omitted in the existing VNS simulations[24,32]. Since they have up to two orders of magnitude smaller diameters than myelinated ones[16,19], and ion channels continuously distributed along the whole axon, the accurate threshold estimation is highly computationally expensive. As a result, there are only either imprecise or computationally inefficient models of C-fibers available[26,33–36]. Therefore, due to computational limitations, past model simulations predicting neural activation considered just large-diameter myelinated fibers, taking only a fraction of the total number of fibers present in a nerve (<1%)[26,37], or employed various techniques to roughly estimate recruitment thresholds instead of simulating the complete fiber dynamics[17,38]. Despite these simplifications, existing models[26,39] still need hours to compute the estimated outcome, even with high-performance computing clusters[40], which drastically limits their use. The mentioned histological and morphological model oversimplifications, together with the limited simulated stimulation paradigms, can provide limited if not misleading results.

While physiological outcomes can be monitored during experiments, precise single-fiber activity cannot be directly measured. Therefore, computational modeling efforts are needed to mimic the relationship between experimentally induced fiber activation and the resulting physiological outcomes. This would enable an understanding of the underlying neural mechanisms causing functional changes, leading to the design of novel treatments. Preliminary efforts have been made to try to predict the recorded compound action potential

(CAP) based on the physiological outcome in rodents[41]. However, these results are hardly transferable to the large animals, due to the significant differences in VN anatomy, since rodents do not have multifascicular VN organization as pigs or humans which have very similar anatomy[31,42].

Developing and experimentally validating computational models that are suitable for use with multiple subjects is a very challenging task. The complexity of personalizing a model to a specific experimental subject is exacerbated by the fact that inter-subject differences are enormous[16,31]. In a study which performed the functional outcome prediction and validation step, models were subject-specific[22]. They are limited to match a posteriori the physiological outcomes on a model developed ex-vivo using the same animal's histology, which makes the entire process not feasible during in-vivo experiments and is possibly subject to biases. Instead, to utilize previously constructed in silico models to optimize stimulation parameters for the new in-vivo animal experiments, they should be personalized for a specific animal using easily measurable and minimally invasive stimulation outcomes.

With the aim to increase the precision of VNS by finding ways to achieve therapeutic effects while minimizing side events, and to understand some of the underlying fiber mechanisms, we developed a histologically and morphologically exact model of the two pigs' VNs. The model replicates nerve composition following fascicles' realistic curvilinear structure and their merging and branching along the nerve length, and implements precisely positioned fibers of all types present in the VN. Thanks to the engineered optimization steps, our model sharply increased the computational efficiency with respect to the state of the art, enabling large-scale simulations. We then designed a method to personalize the model to new subjects during in vivo experiments by matching the model-predicted fiber activation with the laryngeal electromyography (L-EMG), easily measurable during the surgery to accurately replicate the correct nerve-electrode interface, accounting for inherent inter-subject variability and surgical placement. We validated this approach in three new animals using experimentally measured CAP[43], and unveiled which fibers were involved in these mechanisms. As a proof of concept, we tested in silico if the tripolar stimulation can outperform the usually used monopolar stimulation paradigms and reduce unwanted laryngeal muscle activation, enabling further exploration of therapeutic VNS capacity in cardiologic conditions. Our tool holds the potential to guide the design of novel efficient VNS devices and paradigms, while reducing animal experiments.

## Results
### Estimating fiber activation using VaStim−histologically and morphologically realistic model of the cervical vagus nerve stimulation

Starting from the experimental data (Fig. 1a left), we recreated histologically and morphologically realistic model of mid-cervical VN (Fig. 1a right) of two pigs (creating models labeled as M1 and M2). Using four histological cross sections each, spanning a length of 35 mm (Fig. 1a down), we reconstructed the three-dimensional curvature of fascicles, considering their branching and merging. Then, we populated the modeled fascicles with fibers existing in the VN, considering their precise locations, diameters, their type (Aα, Aβ, Aγ, Aδ, B, C) and modality (afferent, efferent), which we labeled using immunochemistry (Fig. 1a).

We thereby created a streamlined end-to-end pipeline (VaStim) able to simulate outcomes of vagus nerve stimulation using arbitrary electrode designs and stimulation policies (Fig. 1b). It exploits the Finite Element Method (FEM) to compute the extracellular potential along all fibers resulting from current injection through electrode active sites. Then, the extracellular potential is applied to neurophysiological models to predict the neural responses of individual fibers[17,26,30,44–49]. As a final outcome, the pipeline estimates which fibers

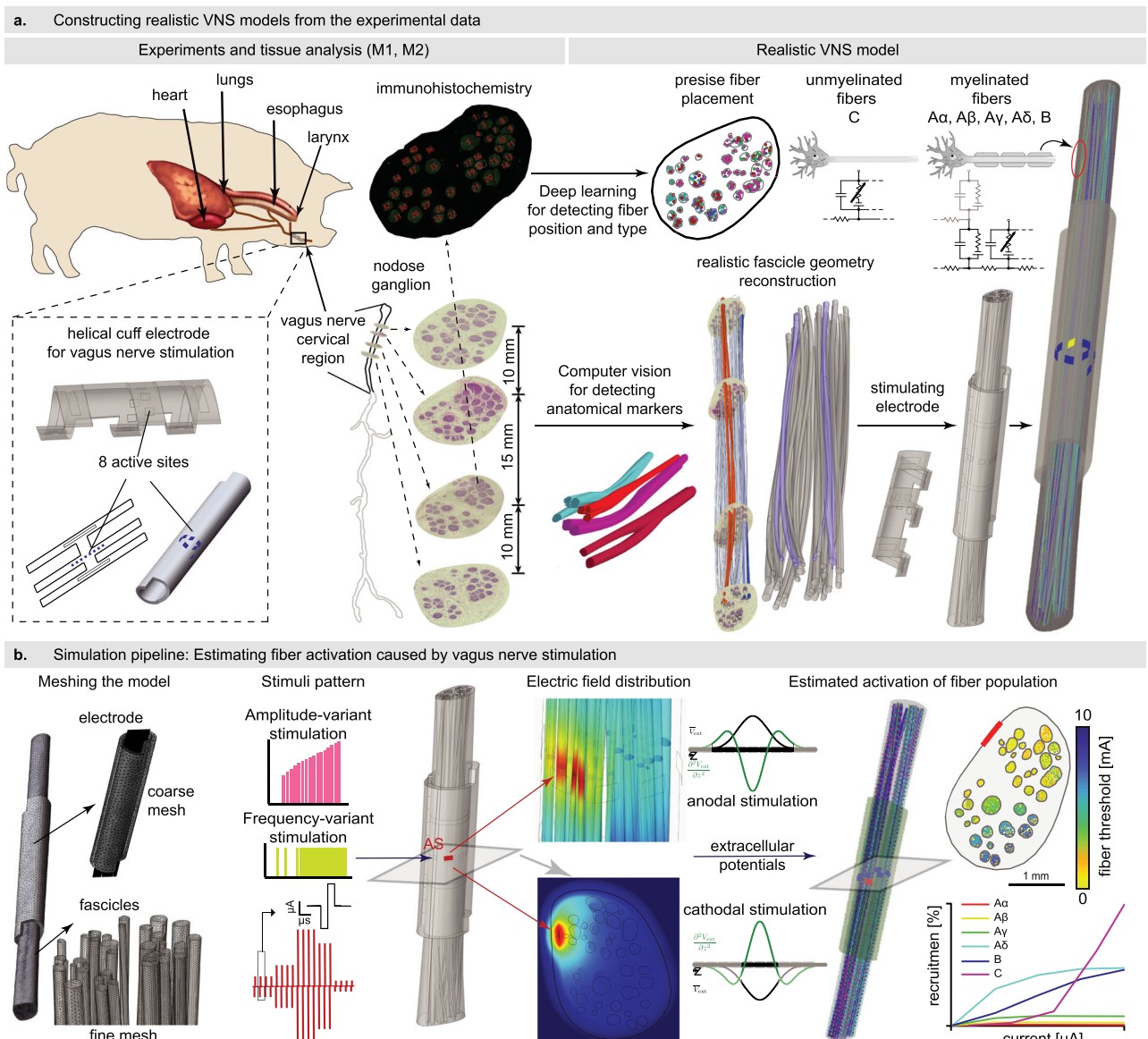

**Fig. 1 | Pipeline for creating and exploiting histologically and morphologically realistic model of cervical vagus nerve stimulation (VaStim). a** The 3D nerve reconstruction is based on histological images of VN cross-sections, considering fascicles' curvatures, branching and merging along the nerve. Immunohistochemistry is applied to determine the precise location, type (Aα, Aβ, Aγ, Aδ, B, C) and modality (afferent, efferent) of different fiber types and the VN is populated with fibers, following fascicle structure along the nerve. We modeled helical nerve cuff that is used in the VNS animal experiments we performed. **b** The nerve structure is meshed. Different stimulation patterns can be applied through electrode active sites (AS) and using numerical solver the resulting voltage is calculated. The examples of resulting electric field distribution are depicted. Resulting voltages are applied as extracellular potentials to multicompartmental axon models, enabling the estimation if VN fibers are recruited or not with the applied stimulation paradigm.

are activated using the applied stimulation paradigm, including their type and location in the VN. It can be used as a tool to replicate experimental results to gain insight on the neural pathways modulating the observed physiological outcomes of VNS, and to evaluate and optimize electrode designs and stimulation paradigms for more effective and safer stimulation.

### Enabling computationally feasible and accurate large-scale simulations of peripheral nerves stimulation

The vagus nerve of the pig contains hundreds of thousands of fibers, majorly unmyelinated[16]. With the aim of estimating the activity of all fiber types across the whole nerve while maintaining reasonable model complexity, we decided to subsample the fiber population. To achieve an optimal tradeoff between the computational cost and resulting accuracy, we performed a convergence study during which we varied the number of uniformly sampled fibers and computed the resulting error in recruitment levels with respect to the complete fiber population, separately for each fiber type (Supplementary Fig. 1). It resulted in a subsample of the fiber population with approximately 10'000 fibers, which limits the mean absolute deviation on recruitment curves below 2%. Final fiber counts are listed in Table 2 and presented in Supplementary Fig. 2.

To assess the fiber activity, we estimate its membrane potential by solving a system of differential equations that models neural dynamics. Numerical solvers require the discretization of the fibers into multiple sections along their length. In case of myelinated fibers, the sections are represented by nodes of Ranvier and internodal compartments[50]. In the case of unmyelinated fibers, which are not physically partitioned by nodes, the discretization is done simply for computational purposes[36]. The membrane dynamics must be solved for each section

at successive, discrete time steps to predict the generation and propagation of action potentials[50,51]. Therefore, computational cost increases with the number of sections as well as time steps. This has rendered previous models unable to perform large scale simulations due to long time needed for computations, often in the order of several seconds for each myelinated fiber[30,46,52], and minutes for each unmyelinated fiber[53], which would result in days or weeks of computations to estimate the response of our fiber population to a single stimulation paradigm. To improve computational efficiency, we developed two methodologies to decrease the number of sections, one general and one specific to unmyelinated fibers (Fig. 2 and Supplementary Fig. 3), and additionally optimized the simulation time step for faster but yet precise simulations (Supplementary Fig. 4).

Unmyelinated fibers are usually discretized using fixed section lengths[26,33,36]. However, this approach is suboptimal. Too coarse discretization length yields inaccurate estimation of recruitment thresholds and, conversely, fine discretization requires significantly more sections to be simulated and therefore much longer computational time. To overcome this efficiency boundary, while maintaining the necessary precision, we developed an algorithm that segments unmyelinated fibers by dynamically assigning section lengths in proportion to the local value of the activation function (AF), corresponding to the second spatial derivative of the extracellular potential[54]. This is inspired by the fact that the AF is a predictor of local depolarization[55]. In comparison to traditional segmentation with uniform section length, this method optimizes the assignment of computational resources to different regions of the fiber based on their relevance regarding the generation of the physiological response. The algorithm therefore discretizes a fiber so that regions where the magnitude of the AF is large are resolved finely (using smaller sections), whereas regions with a smaller magnitude are resolved more coarsely (using larger sections) (Fig. 2a).

Employing dynamic discretization allowed to drastically reduce the simulation time per unmyelinated fiber, from $2807.7 \pm 389.7$ ms to $334.5 \pm 17.7$ ms, which equals to a speedup factor of $8.4 \pm 1.4$ (Fig. 2b and Supplementary Table 1). The resulting deviation of fiber recruitment thresholds with respect to fixed-length discretization was minimal ($0.4 \pm 1.6\%$, Fig. 2b and Supplementary Table 1). The difference was larger than 5% for only 51 (0.08%) of the data points, therefore our method is substantially maintaining its validity.

The extracellular potential decays to zero away from the location of the stimulating electrode. Therefore, the end regions of nerve fibers, further to the current source, are less relevant for accurately predicting a fiber's physiological response. We therefore avoided unnecessary computations by truncating the fibers so that nodes at the extremities, whose normalized extracellular potential is lower than a certain truncation threshold, are not simulated (Fig. 2c and Supplementary Fig. 3a). We determined that the precise truncation threshold (Fig. 2c and Supplementary Fig. 3b) depends on the fiber diameter and type (myelinated or unmyelinated). We evaluated it a-priori by conducting a study on a sample set of fibers, whose diameters are assigned from a discrete set of values. Longitudinal truncation reduced computation time by $13.7 \pm 1.2$ times for unmyelinated fibers and for myelinated fibers (from $1.2 \pm 0.3$ for Aα to $8.2 \pm 1.0$ for Aδ), introducing relative errors well below 0.5% (Fig. 2d, Supplementary Fig. 3b and Supplementary Table 1).

The numerical solver computes the evolution of membrane potential iteratively at each discretized time step. Therefore, the computation time is proportional to the number of time steps and thus to the resolution of discretization. A standardly used time step value for both the myelinated axon model (McIntyre, Richardson, and Grill-MRG) and unmyelinated axon model (Tigerholm-TH) is 5 μs[33,36,37,50]. We increased the time step while simultaneously computing the resulting error of predicted thresholds until the deviation remained below 1.5% (Supplementary Fig. 4). This procedure resulted with the optimal time

step of 13 μs. Compared to the previously used time step of 5 μs, it reduced simulation times by $2.8 \pm 0.5$ for the MRG and $2.6 \pm 0.8$ for the TH model. In Supplementary Table 1 the resulting computational speedup and relative deviation of recruitment thresholds are compiled by fiber type ($n = 1694$ for MRG; $n = 478$ for TH).

We finally evaluated the improvement of computational efficiency achieved by combining methods previously presented. We calculated the speedup and the introduced error of the optimized model in comparison to a baseline model with fixed discretization of unmyelinated fibers, without longitudinal truncation, and by using the time step of 5 μs. All optimizations combined provided a reduction of computational time: $117.3 \pm 18.7$ times reduced for unmyelinated fibers, and between $2.8 \pm 0.8$ (Aα) and $17.3 \pm 2.5$ (Aδ) times reduced for myelinated fibers. The relative errors introduced were between $0.5 \pm 1.6\%$ (C fibers) and $2.5 \pm 3.0\%$ (Aα fibers) (Fig. 2e, Supplementary Fig. 3c and Supplementary Table 1). The relative deviations of recruitment thresholds are larger than 5% for just 0.08% of the C fiber data points.

## Nerve models considering realistic fascicular branching and curvilinear structure yield significantly different predictions with respect to simplified extruded models

Models developed up-to-date[26,38,44,46,49] typically assume the fascicles to be straight tubes, simply extruded from a single histological image taken at a certain level. We investigated to what extent the histologically accurate modeling of three-dimensional fascicle branching and merging influences model predictions compared to linearly extruded fascicles. Starting from the segmentation of a longitudinal sequence of cross sections, we created anatomically plausible nerve models, considering also fascicles' curving, merging and branching. On the other hand, we generated simplified models as linear extrusion from an intermediate cross-section. Then, we placed the electrode in three locations on both curvilinear and extruded models (Fig. 3a ii). We evaluated the relative deviation of each fiber recruitment thresholds estimated with curvilinear and extruded models for each electrode location (Fig. 3a iii). We observed significant differences ($p < 0.0001$) between thresholds estimated by curvilinear and extruded models already with the electrode placed at a distance of 2.5 mm from the extrusion level ($2.4 \pm 13.0\%$, $t(74282) = 49.5$ for M1, $-1.4 \pm 11.0\%$, $t(77055) = -35.7$ for M2). Therefore, respecting the realistic 3D development of the nerve is necessary to properly estimate the nerve response to stimulation since large errors are committed even when the extrusion is performed very close to the stimulation site. Moreover, the deviation sharply increases when the electrode is placed farther from the extrusion level, with a significant bias ($p < 0.0001$) which renders inaccurate even population-averaged results ($13.8 \pm 28.7\%$, $t(74282) = 131.0$, and $22.6 \pm 44.5\%$, $t(74282) = 138.1$ for M1, and $7.6 \pm 23.1\%$, $t(77055) = 90.6$, and $22.5 \pm 36.7\%$, $t(77055) = 169.9$ for M2, respectively for distances of 7.5 mm and 12.5 mm from the extrusion level toward the thoracic region).

## Precise fiber placement is important for the development of selective stimulation policies

Axons tend to group by type and modality at the cervical level of vagus nerve[16]. We investigated whether incorporating the grouping of fibers in the model (Fig. 3b i) can facilitate designing more selective stimulation policies. Using our experimental data, we first quantified the magnitude of clustering of individual fiber groups using the silhouette coefficient[56] (Supplementary Fig. 5), and we observed high level of grouping of $B_{Eff}$ and $C_{Eff}$ fibers, as well as of $A_{Eff}$ in M1 (Fig. 3b i and Supplementary Fig. 5a, b). Then, we evaluated the effect of clustering on possible achievable selectivity. We developed anatomically identical models, which we populated with the same population of fibers as the histologically accurate models, but randomly placed them across all fascicles, neglecting the existence of clustering. We calculated the

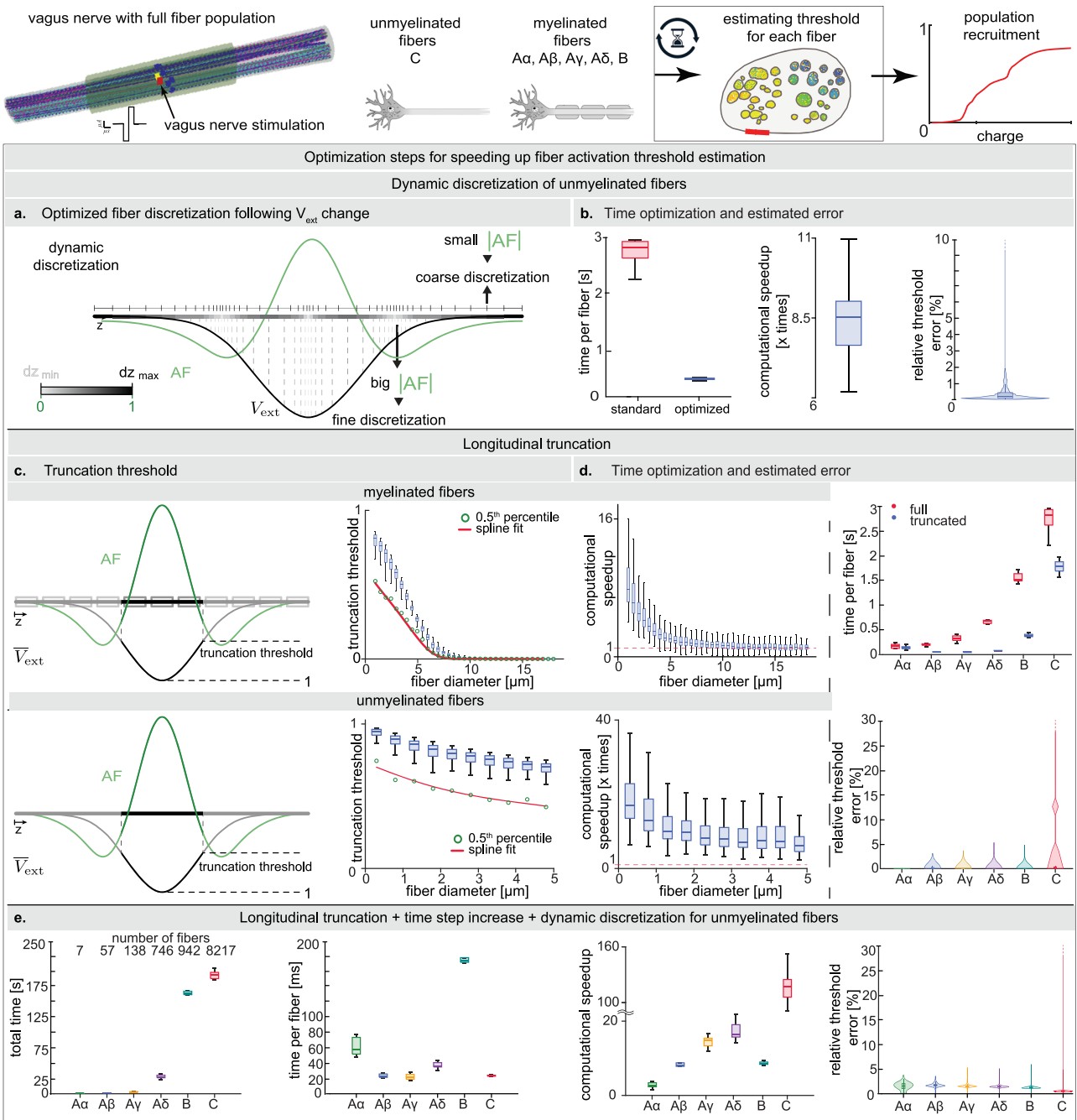

**Fig. 2 | Optimization methods to increase the time efficiency for fiber activation threshold estimation.** All simulations are performed with a cathodic pulse of 500 μs width from each of the eight active sites. **a** Illustration of the dynamic discretization algorithm for unmyelinated fibers. The length discretization step is proportional to magnitude of the activation function (in green). **b** Simulation time per unmyelinated fiber (mean per active site), resulting computational speedup (mean per active site), and relative deviation of recruitment thresholds when using dynamic discretization ($n = 65{,}736$: 8217 unmyelinated fibers per eight active sites). **c** Illustration of longitudinal truncation for myelinated (top) and unmyelinated (bottom) fibers. Sections at the extremities not relevant for predicting the physiological response, i.e., with AF below a certain threshold, are not computed (grayed out on figure). On the right, the truncation thresholds determined by a study conducted for myelinated (top) and unmyelinated (bottom) fibers ($n = 368$ to 400 fibers for each diameter class, 17,376 in total). **d** Computational speedup when using longitudinal truncation for myelinated (top left) and unmyelinated (bottom left) fibers ($n = 368$ to 400 fibers for each diameter class, 17,376 in total). On the right, the related simulation time per fiber (mean per active site, $n = 8$ per fiber type) and the relative deviation of recruitment thresholds for all fibers ($n = $ Aα: 7; Aβ: 57; Aγ: 138; Aδ: 746; B: 942; C: 8217 fibers, per eight active sites). **e** Total simulation time per active site for all fibers of each fiber type, simulation time per single fiber (mean per active site), computational speedup per fiber type, (mean per active site) and relative deviation of recruitment thresholds, with all presented methods combined ($n = $ Aα: 7; Aβ: 57; Aγ: 138; Aδ: 746; B: 942; C: 8217 fibers, per eight active sites). In (**b**–**e**), boxplot elements are defined as follows; center line: median; box limits: upper and lower quartiles; whiskers: 1.5x interquartile range.

maximum achievable selectivities (calculating the selectivity obtained for the fiber group and taking the maximum across active sites) and compared them with the ones obtained with histologically and morphologically realistic model (Fig. 3b ii and Supplementary Fig. 5c).

The best achievable selectivity for each fiber type increases when the precise spatial organization of fibers is considered in the model, particularly for $B_{Eff}$, $C_{Eff}$, and $C_{Aff}$ fibers. To further highlight the significance of this finding, we complied the recruitment curves of $B_{Eff}$

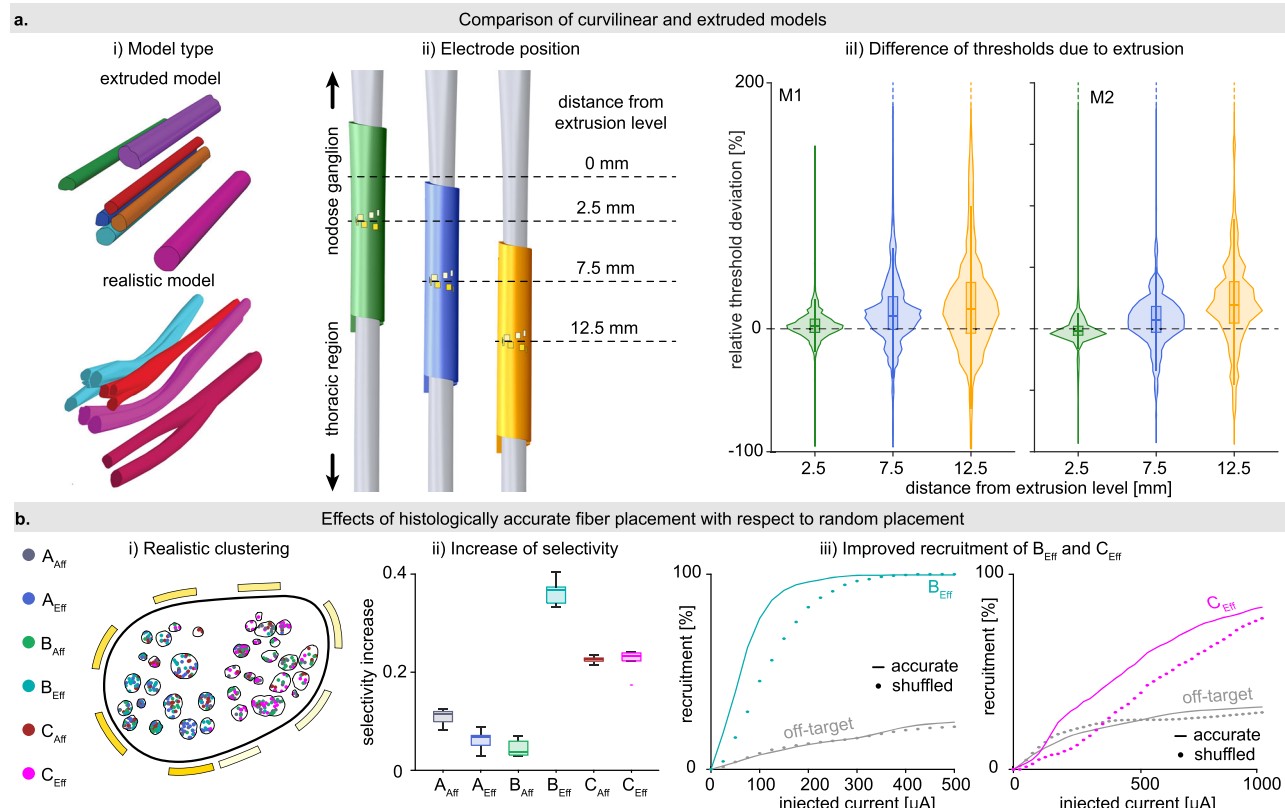

**Fig. 3 | Comparison of histologically and morphologically realistic and simplified nerve models. a** Comparing recruitment threshold estimates between models with anatomically plausible and linearly extruded fascicles. i. Depiction of linearly extruded and anatomically plausible reconstructions of selected fascicles. The anatomically plausible model considers three-dimensional fascicle propagation and their merging and branching. ii. Illustration of the three electrode placements for which the recruitment thresholds were compared, with distances from the extrusion level being indicated. iii. Distributions of the relative recruitment threshold deviations grouped by electrode placement and labeled by their distance from the extrusion level ($n = 74{,}296$ for M1, $n = 77{,}056$ for M2). A trend of increase of deviations toward the thoracic region can be observed for both nerve models, which corresponds to a decreasing trend of recruitment thresholds from nodose to thoracic regions. Boxplot elements are defined as follows; center line: median; box limits: upper and lower quartiles; whiskers: 1.5x interquartile range. **b** Comparing

the achievable selectivity of specific fiber types for histologically accurate and spatially shuffled fibers. i. Cross section of the nerve model with the locations of sampled fibers colored by type (equal counts per type). ii. The maximum increase in selectivity for the most selective active site per fiber group ($n = 8$ per fiber type), within the stimulation range (0–1 mA). Boxplot elements are defined as follows; center line: median; box limits: upper and lower quartiles; whiskers: 1.5x interquartile range; points: outliers. iii. The recruitment curves for $B_{Eff}$ and $C_{Eff}$ target fiber groups for their most selective active sites together against the recruitment curves of off-target fiber groups. The recruitment curves of the model with histologically accurate fiber placements (solid lines) show that clustering allows for more selective stimulation relative to off-target recruitment, compared to the model with randomly shuffled fibers (dotted lines). The results shown in this panel refer to M1.

and $C_{Eff}$ for the most selective active sites as well as the corresponding recruitment curves of the remaining (off-target) fibers (Fig. 3b iii and Supplementary Fig. 5d). As the stimulation current increases, the activation of the target population is higher in histologically accurate models than in the shuffled models, whereas the off-target recruitment remains comparable, which corresponds to a considerable improvement of selectivity. Placing the fibers according to their real spatial organization showed to be crucial to properly predict the achievable selectivity and can be exploited in real-life scenarios to improve the selectivity for fiber types which are responsible for the desired clinical outcomes.

## Models match well the experimentally obtained recordings after personalization

To employ the developed models in-vivo on new experimental subjects and match physiological responses with the predicted fiber activation, we developed a methodology for personalizing the models using laryngeal electromyographic recordings (L-EMG) of the thyroarytenoid muscle (Fig. 4a), easily measurable during the surgery. This method is based on the fact that these large fibers are the first to be recruited by electrical stimulation[55,57,58] and the assumption

that activation of Aα efferent fibers induces the laryngeal muscle response[13,16,22,41,59–61]. We rotate the electrode with the modeled active sites (thanks to the axial symmetry of the cuff electrode) to mimic possible variations during surgical placement, and scale model-estimated thresholds to account for individual variability at the nerve-electrode interface. We applied this method to personalize both M1 and M2 to three new experimental subjects (S1, S2, and S3). After applying the personalization method, the recruitment curves of Aα efferents have shown a high predictive power for the experimentally measured L-EMG, reaching an R2 of $0.77 \pm 0.07$ (Fig. 4b). Figure 4b illustrates an example of the similarity between the personalized model and experimental recordings, in terms of recruitment per active site. Results of personalization for each model-experimental subject pair (3 subjects, 2 developed models) are presented in Supplementary Fig. 6a. When considering all pairs ($n = 6$), the personalized models showed significantly higher performance compared to the models prior to personalization ($p = 0.031$).

We validated the models by comparing the model estimated recruitment curves of Aα and Aβ efferent fibers with the corresponding amplitude of compound action potential (CAP) of fast

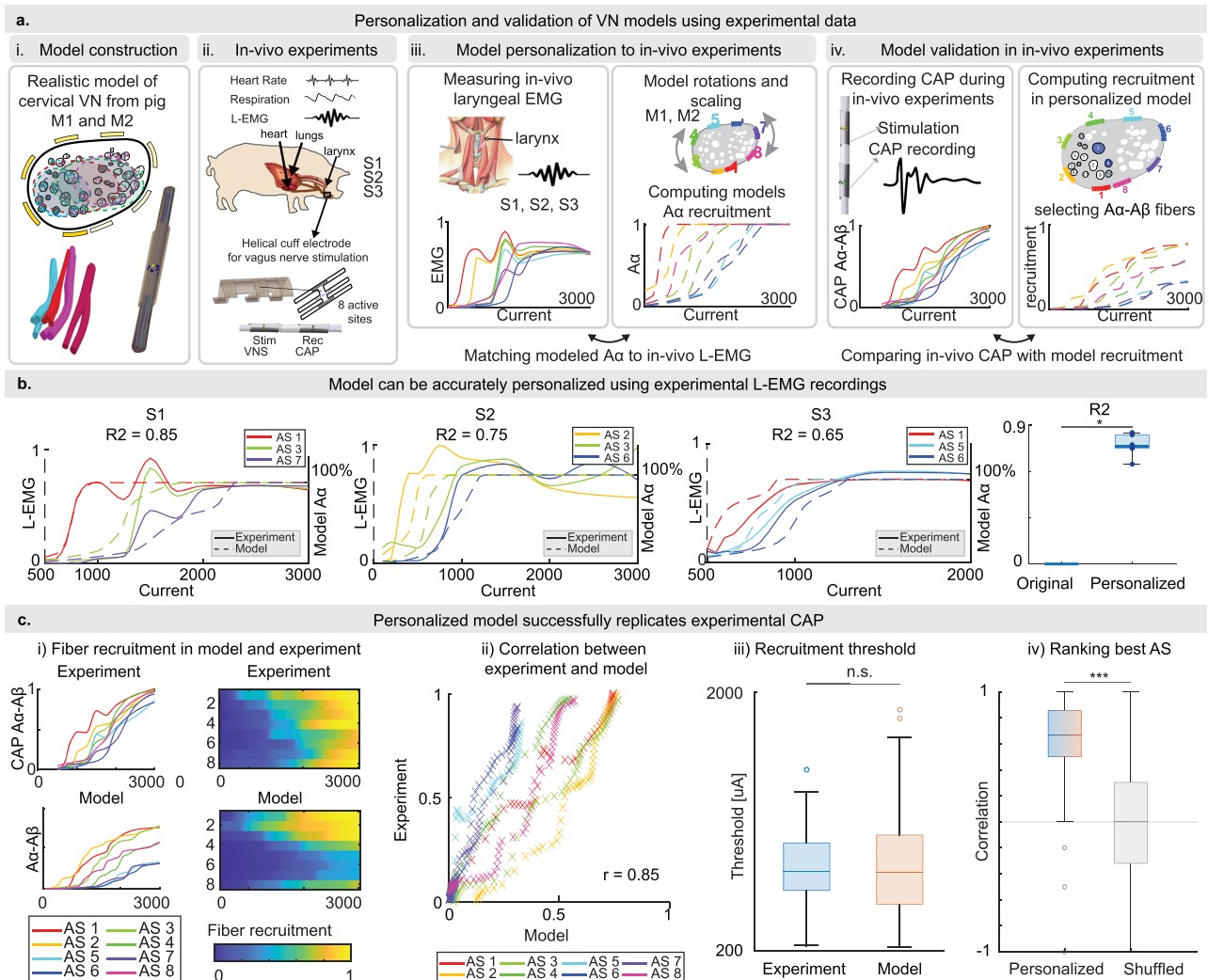

**Fig. 4 | Model personalization and validation. a** Experimental timeline and its steps: i. Construction of a realistic vagus nerve model from M1 and M2. ii. Experimental setup for the previously unseen subjects S1–S3. iii. Personalization of the models to the new experimental subjects, including the simple L-EMG recording setup, and customization of the model by rotation of active sites and scaling of thresholds. iv. Validation using CAP. **b** On the left, recruitment curves of M1 personalized to the three experimental subjects (S1, S2, S3), overlaid to the experimentally measured L-EMG, for three active sites. Results for all personalized models are presented in Supplementary Fig. 6a. On the right: R2 comparing personalized vs. original linear regression between models and experimental subjects ($n = 48$, 6 pairs by 8 active sites). Boxplot elements are defined as follows; center line: median; box limits: upper and lower quartiles; whiskers: 1.5x interquartile range; points: single samples. **c** i. In the first column, first row the normalized Aα and Aβ CAP curves are reported per active site for S1. The corresponding recruitment curves of Aα and Aβ estimated by the personalized M1 are reported in the second row. The second column represents the same data as heatmaps. ii. The normalized CAP from S1 and M1 model-predicted recruitment level for each active site and current level are correlated on a scatter plot. Pearson's correlation coefficient is reported. Results for all personalized models are presented in Supplementary Fig. 6b. iii. The distribution of thresholds to recruit 10% of Aα and Aβ fibers in all personalized models are compared to the experimentally measured thresholds to obtain a 10% of CAP value (two-sided Wilcoxon signed rank test, $p = 0.83$, $n = 48$: 8 active sites per 6 personalized models). iv. The ranking of active sites predicted by all personalized models is compared to the experiment by computing Spearman's correlation on thresholds, for recruitment levels between 10% and 70% (two-sided Wilcoxon signed rank test, $p < 0.0001$). In the left box, it is reported the distribution of correlations for the personalized models ($n = 48$: 8 active sites per 6 personalized models). In the right box, it is reported the distribution of correlation values obtained when shuffling the order of active sites, repeated 100 times ($n = 4800$: 8 active sites per 6 personalized models, per 100 randomizations). Boxplot elements are defined as follows; center line: median; box limits: upper and lower quartiles; whiskers: 1.5x interquartile range; circles: outliers.

(large diameter) fibers recorded from cuffs implanted caudally to the stimulation electrode[16] (Fig. 4c i and Supplementary Fig. 6b i). The model-predicted and experimental curves showed high correlation (0.88 ± 0.03) (Fig. 4c ii and Supplementary Fig. 6b ii). The amount of current necessary to recruit 10% of Aα-Aβ in the model was not significantly different from the thresholds to obtain a 10% recorded value of CAP ($z = 0.22$, $p = 0.83$, $n = 48$, $r = 0.03$) (Fig. 4c iii and Supplementary Fig. 6b iii). Additionally, the models exhibited good predictive power when ranking the best active sites, determining which active sites had lower or higher thresholds compared to the others, as indicated by Spearman's correlation coefficient of 0.67 ± 0.17 ($n = 42$). The

distribution of correlations significantly differed from the correlation obtained after shuffling the order of active sites (0.00 ± 0.38, $n = 4200$, $z = 8$, $p < 0.0001$, Fig. 4c iv).

## Models help to understand the mechanisms of fiber activation for heart and laryngeal function and can guide the design of more selective stimulation

The aim of VNS in cardiovascular applications is to modulate the heart activity, potentially treating heart failures, arrhythmias, and other cardiological issues[9,10,62,63]. Exploring VNS beneficial effects is often limited by the fact that prior to inducing relevant heart activation

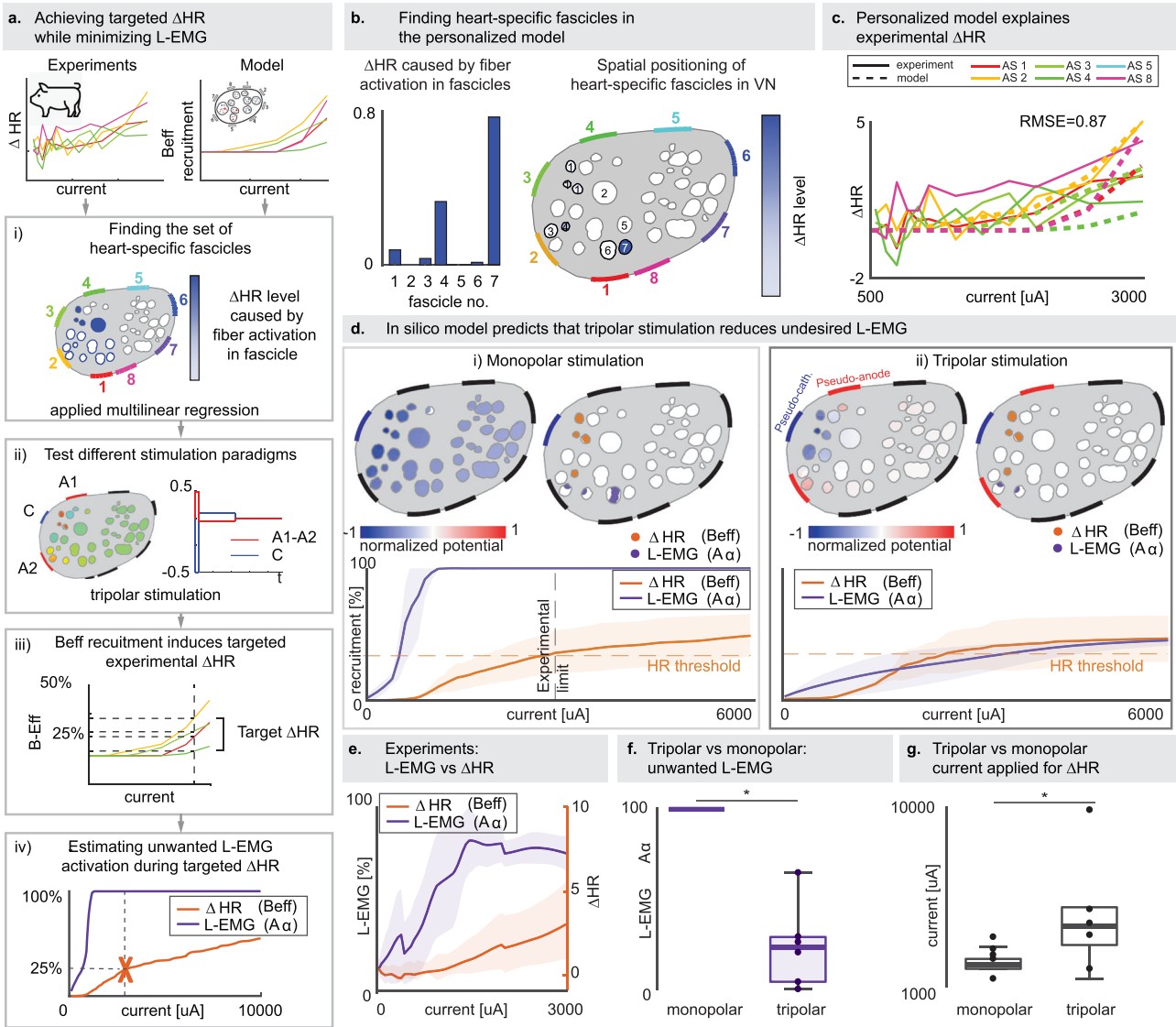

**Fig. 5 | Explanation of experimental heart rate variation through in silico models and improvement of selectivity through tripolar stimulation. a** Pipeline to optimize HR changes while minimizing L-EMG side effects. Fascicles of the personalized models that are responsible for experimental HR changes are extracted. Then different stimulation paradigms can be tested on the personalized models, and HR vs. L-EMG recruitments are computed. **b** Finding heart-specific fascicles M1 personalized to S1. Level of HR change (left, bars) caused by fiber activation in specific fascicles. Fascicles are spatially presented and color-coded based on these values (right). Results for all personalized models are reported in Supplementary Fig. 7a. **c** Recruitment curves of M1 personalized to S1, overlaid to the experimentally measured ΔHR, for active sites with recruitment higher than 0 in the experimental range. Results for all personalized models are reported in Supplementary Fig. 7b. **d** Top part: example of monopolar (left) vs. tripolar (right) stimulation. For each stimulation paradigm the normalized potential and the ΔHR

$B_{Eff}$ and L-EMG Aα activations are shown. Bottom part: the off-target (Aα) recruitment and target ($B_{Eff}$) recruitment between monopolar and tripolar case. **e** Experimental recruitment curves of ΔHR and L-EMG. The solid line in (**d**, **e**) represents the mean, shaded areas ±1 SD, across all model-experimental subject pairs ($n = 6$). **f** Comparison of monopolar vs. tripolar off-target (Aα) recruitment at 25% recruitment of $B_{Eff}$ (model-estimated fiber activation at experimentally obtained threshold stimulation amplitude, causing clinically relevant ΔHR) for all pairs model-experimental subject (two-sided Wilcoxon signed rank test, $p = 0.031$, $n = 6$ personalized models, $z = 2.2$). **g** Comparison of monopolar vs. tripolar charge to recruit 25% of $B_{Eff}$ (targeted ΔHR) for all pairs model-experimental subject (two-sided Wilcoxon signed rank test, $p = 0.031$, $n = 6$ personalized models, $z = -2.2$). In (**f**, **g**), boxplot elements are defined as follows; center line: median; box limits: upper and lower quartiles; whiskers: 1.5x interquartile range; points: single samples.

changes, subjects are experiencing coughing or breathing problems. Therefore, our objective was to obtain the high selectivity of heart modulation with minimal off-target stimulation of L-EMG. We hypothesized that the activation of the whole population of Aα efferent fibers induces the laryngeal muscle response[13,16,22,41,59−61], which we experimentally measured. On the other side, we assumed that the reduction in heart rate, which we recorded during experiments, was due to the activation of only a subset of $B_{Eff}$ fibers[13,16,22,41,59−61]. Therefore, in models personalized for each animal, we identified heart-specific fascicles. To do so, we excluded the ones with a low number of

$B_{Eff}$, identified through 2-means clustering, and those that did not exhibit any $B_{Eff}$ activation within the experimental current range. We then fitted a linear regression model (see Methods) where the experimentally measured reduction in heart rate (ΔHR) was predicted with the amount of $B_{Eff}$ recruited for each selected fascicle. Using this method, we assigned a "functional weight" to each selected fascicle (Fig. 5b), representing the importance of their activation in heart rate modulation. The RMSE between model-predicted and experimentally measured heart rate variation was $1.04 ± 0.23\%$ (Fig. 5c). Results from the same method for both M1 and M2 personalized to

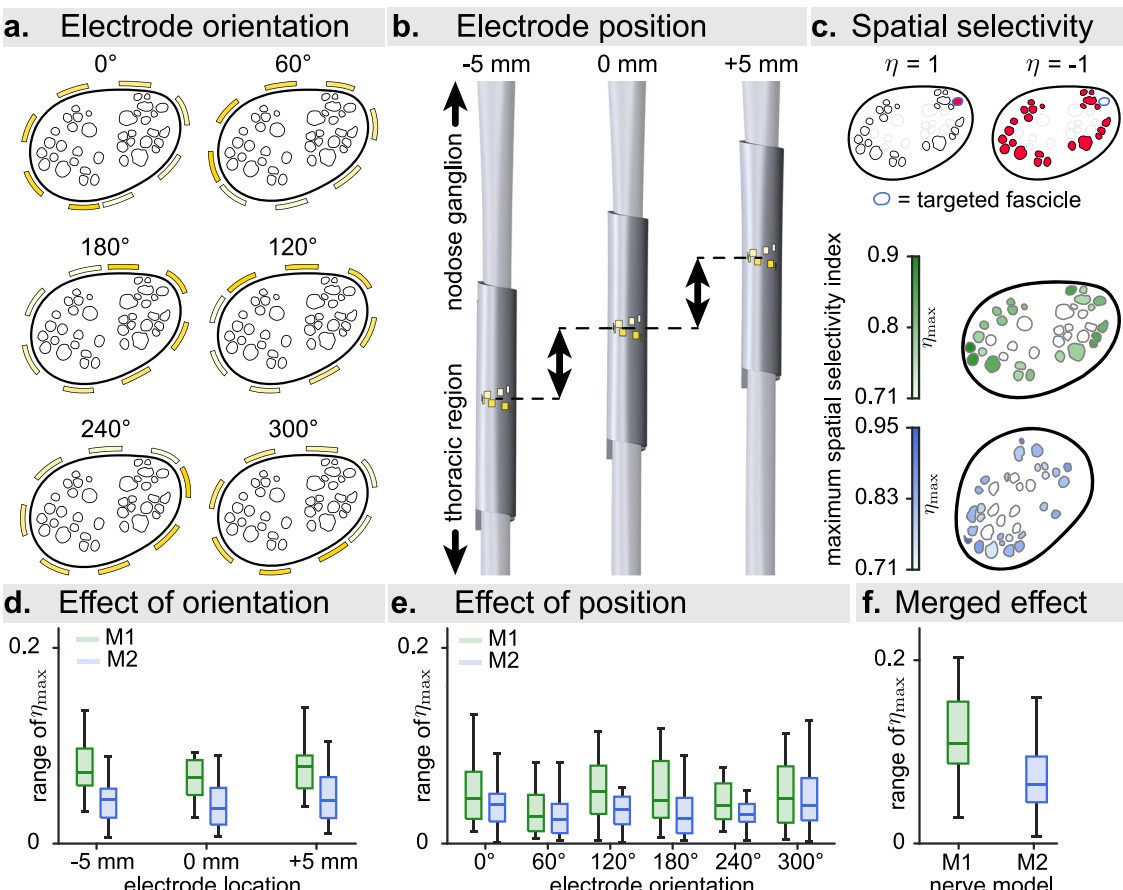

**Fig. 6 | Precise electrode surgical placement is not influencing maximum achievable functionality.** Range spanned by best spatial selectivity index due to rotation and translation of the helical cuff electrode. **a** Illustration of the six considered electrode orientations. **b** Depiction of the three considered electrode locations. **c** Above: representation of the two extreme cases of the spatial selectivity index $\eta \in [-1,1]$. The index is calculated for a targeted fascicle (contoured in blue). Red fascicles indicate recruitment of all contained fibers. $\eta = 1$ corresponds to the activation of all fibers in the targeted fascicle without the recruitment of any other fiber, $\eta = -1$ to the activation of all fibers in all fascicles except the targeted one (see

Eq. (8)). Below: Maximum spatial selectivity $\eta_{max}$ achieved for each fascicle over all active sites when varying the stimulation amplitude. Fascicles with $\eta_{max} < 0.7$ were removed. **d** Range of $\eta_{max}$ over all electrode orientations per electrode location ($n = 54$ per boxplot, 6 orientations by 9 fascicles). **e** Range of $\eta_{max}$ over all electrode locations per electrode orientation ($n = 27$ per boxplot, 3 locations by 9 fascicles). **f** Range of $\eta_{max}$ over all electrode orientations and locations ($n = 162$ per boxplot, 6 orientations by 3 locations by 9 fascicles). In (**d**–**f**), boxplot elements are defined as follows; center line: median; box limits: upper and lower quartiles; whiskers: 1.5x interquartile range.

each experimental subject, and for all active sites, are presented in Supplementary Fig. 7.

To evaluate the selectivity of the chosen stimulation paradigm, we compared the experimental curves of ΔHR and L-EMG in three pigs. These curves revealed a rapid increase of L-EMG activation at low current values and a much slower increase in ΔHR, which became clinically relevant[16] only toward the end of the stimulation range, after saturation of L-EMG (Fig. 5e). This suggests that, with the chosen electrode and stimulation paradigm, the selectivity for HR reduction is strongly limited. This behavior is accurately replicated in the models, where the recruitment curves of Aα saturate much faster than the recruitment curves of $B_{Eff}$ (Fig. 5d i) in the fascicles identified as the ones responsible for ΔHR (Fig. 5a i). We then used the model to assess the performance of a more complex stimulation paradigm aimed at enhancing the selectivity for ΔHR while reducing off-target L-EMG. Using the models, we estimated the extent of $B_{Eff}$ recruitment induced by the stimulation amplitude, defined as the threshold required for achieving the targeted heart rate change[16]. By averaging the value obtained from models personalized for each animal, we assumed that 25% of $B_{Eff}$ recruitment could be considered as relevant target. We first identified the active site with the lowest threshold to obtain 25% recruitment of the target $B_{Eff}$ in the monopolar case, which corresponds to complete Aα saturation (100% recruitment) as observed in

experiments (Fig. 5d i). We then performed a new simulation where we stimulated the nerve in a tripolar configuration with an asymmetric waveform, where the previously identified active site served as the pseudo-cathode, and the active sites on either site were set as pseudo-anodes (Fig. 5a ii, d). Finally, we compared the off-target (Aα) recruitment at the 25% target ($B_{Eff}$) recruitment between the monopolar and tripolar cases, revealing a significant reduction in off-target activation using the tripolar paradigm (68% ± 21%, Fig. 5f), at a cost of higher injected current (ratio of 1.9 ± 0.9, $p = 0.031$, Fig. 5g).

## Electrode performance is robust to small variations in surgical placement

Wrapping the electrode around the nerve at a precise location and with a specific orientation is very challenging during the surgical procedure, due to the absence of adequate visual landmarks. We investigated whether this variation in surgical placement effects the performance of the electrode. We applied the cuff on both M1 and M2 with six different (rotational) orientations and at three different (longitudinal) locations along the nerve, which spanned a range of 10 mm (Fig. 6a, b). We then evaluated the variation of selectivity (Fig. 6c) due to the influence of electrode rotation (Fig. 6d), translation (Fig. 6e), the combination of both (Fig. 6f). We observed that the maximum variation of selectivity was always below 0.1, concluding that the overall performance of the

helical cuff electrode does not considerably depend on the exact surgical placement, i.e., different surgical placements can be compensated by a choice of different active sites for the stimulation.

## Visualizing models and simulation results using a web-based platform

We developed a web-based platform to visualize the models and simulation results with a fully developed graphical interface, requiring only an internet browser. It is publicly accessible[64] and showcases, for both nerves, all models and electrode placements which were examined in this paper. The platform allows to manipulate the 3D model view, which comprises the geometries of epineurium, fascicles, and electrode as well as the whole fiber populations, color coded by type. Furthermore, the considered stimulation policy and resulting recruitment curves can be plotted.

## Discussion

In this study, we developed a histologically and morphologically realistic computational model of vagus nerve stimulation, VaStim. It is personalized and validated using experimental data, and then exploited in combination with animal experiments. To do so, we reconstructed the three-dimensional propagation of fascicles, considering their curvatures, as well as merging and branching along the nerve, instead of commonly used linear extrusions[21,24–27]. We then populated the nerves with individual fibers of all types, whose location, type, modality, and diameter were obtained through single-axon immunohistological analyses, considering also often omitted, computationally demanding fibers, thanks to computation optimization methods that we engineered. We designed a method to personalize the model for new in-vivo animal experiments and validated it using data from experiments performed with three new pigs. We developed a mechanistic model-based understanding of the interplay between fibers involved in neuromodulation of the heart and of the thyroarytenoid muscle, measured experimentally, and proposed the future method to enhance the therapeutic viability of VNS. In essence, VaStim is an in silico tool that can predict fiber activation and its functional outcomes by applying an arbitrary stimulation policy through electrode of any design.

## Computationally efficient model can accurately simulate the behavior of large scale of myelinated and unmyelinated fibers in the vagus nerve

We engineered and implemented several methodologies (longitudinal truncation, time step increase, and dynamic discretization) to reduce the time necessary for threshold calculation per fiber and therefore optimize the overall computational efficiency compared to currently available models. Indeed, previous models were used to simulate only a small number of fibers[26], or fiber populations excluding unmyelinated fibers[30,48].

Unmyelinated fibers are particularly difficult to simulate due to the fine spatial discretization necessary to estimate their response accurately. As a result, either approximate or computationally heavy models of C-fibers exist[33–36]. To overcome this issue, we introduced a dynamic discretization algorithm that optimally partitions the axons so that the resolution of discretization is proportional to the AF for each location along the fiber. This method ensures precision in the areas where the action potentials are generated, while it decreases the computational cost by reducing the resolution in unaffected areas.

Longitudinal truncation excludes from the simulation portions of the fibers which are not relevant for predicting an accurate physiological response, both for myelinated and unmyelinated fibers. Previously, myelinated fibers have been simulated for a fixed number of nodes[44,50,65], however, our results show that the number of nodes which must be modeled to obtain a reliable prediction depends on the axon diameter and stimulation paradigm (Fig. 2c and Supplementary

Fig. 3a). The method we propose is able to optimize the use of computational resources while maintaining control over the error for all fiber diameters and stimulation paradigms. Finally, an optimization of the time discretization step allowed to additionally reduce computational times for all fiber types.

These steps significantly reduced the computation cost, especially for unmyelinated small diameter C-fibers. We demonstrated that by using VaStim it is feasible in short time spans to execute large-stale studies of vagus nerve stimulation, containing more than 10 thousand fibers of all types for accurate estimation of recruitment, and up to the whole fiber population of the vagus nerve when the activity of each single axon is of interest. Previous models implementing unmyelinated fibers simulated a much smaller number of fibers such as one per fascicle[21]. Ultimately, we decreased the simulation time required for such a study from more than 7 h to 6.7 min (Supplementary Table 1) while using a 48-cores of a high-performance cluster. The time necessary to perform these simulations is reduced by multiprocessing since the computation of single fiber responses is an embarrassingly parallel problem. However, since the computational benefit is less than linear with the number of available processing threads[66], the simulations would also be performed in a reasonable time span on a standard quad-core desktop computer. We achieved this efficiency optimization while obtaining very accurate results. Indeed, we observed deviations of recruitment thresholds by <2.5% for myelinated and <0.5% for unmyelinated fibers, negligible in most usage scenarios. Importantly, our methodology can enhance the performance of computational modeling in other fields of neurostimulation, as our optimization strategy can be generalizable to any other nerve models of this kind[26,37,46].

## Morphologically and histologically realistic model of VNS enhances the precise prediction of activated fibers

Analysis of experimentally obtained morphological data of VN revealed that fascicles are twisting, merging and branching along the rostro-caudal axis[16,28,29]. However, the majority of published 3D nerve models have been constructed by linearly extruding fascicular structures (epineurium and endoneurium)[17,23,24,26,46,67]. In our previous studies on different peripheral nerves, we explored the integration of anatomically plausible propagation and branching of fascicles[30,37,48]. While incorporating a curvilinear fascicle structure considering merging and branching along the rostro-caudal axes is more realistic, we also quantified its practical importance. To assess this, we compared the thresholds predicted by our morphologically and histologically realistic models to those estimated by linearly extruded models, both populated with identical fiber distributions. We observed that large relative deviations already when the extrusion is performed at the level of stimulation, with a standard deviation above 10%. Moreover, we observed a bias which increased when the electrode is moved toward the thoracic region, surpassing 20% when the stimulation is performed just 12.5 mm away, most possibly due to the longitudinal variation in the nerve cross-section. Indeed, within the anatomically realistic model, where the epineurium thickness varies along the nerve, the distance of the wrapped active sites to the fibers also varies. In contrast, the linearly extruded model maintains a constant epineurium thickness. Therefore, the positive error trend observed is likely associated with the thickening of the epineurium toward the nodose region, properly represented in the realistic model. Consequently, linearly extruding fascicles yields significantly different thresholds compared to morphologically accurate modeling.

Previous computational models of nerves were commonly populated with uniformly distributed fibers among fascicles[26,30,37], discarding possible existence of clusters. Instead, we integrated precise fiber data obtained from single axon immunohistological analyses into our model. To understand the impact of considering this realism on simulation predictions, we compared the results obtained from

models with histologically accurate fiber placements to those where the same fibers were uniformly shuffled across all fascicles in the respective cross section. Throughout this investigation, we observed the significant influence of naturally occurring clustering of fibers within the vagus nerve on the best achievable selectivity, particularly prevalent for efferent type B and C fibers. Therefore, uniformly distributed fibers could lead to the incorrect choice of the most selective active site or optimal charge injection. Furthermore, considering that different physiological functions are controlled by different fiber types located in different fascicles[13,16,22,41,59–61], accurate fiber clustering in the model becomes crucial for enhancing the precision of predictions regarding expected physiological changes.

## Personalized and validated computational models can be exploited for understanding functional outcomes and testing new stimulation paradigms

Experimental evidences reveal significant inter-subject variability in vagus nerve anatomy[16,31]. Therefore, it is expected that the computational models need to be adapted before using it for simulating the VNS outcomes of a new subject. We have developed a method to personalize the models for a new animal undergoing experimental VNS testing. Previous models, lacking this step, could only be used *a posteriori*, i.e., at the end of the experimental protocol after nerve explant[30,47]. By using very easily measurable laryngeal muscle recordings, the personalization step tunes the model for the anatomy of the specific animal. First, as the cuff electrode is axially symmetric, it rotates the order of modeled active sites, to align the active sites to the precise placement of the cuff during the surgery. Then, it scales the model-estimated thresholds to represent individual variability at the interface between nerve and electrode[30,47]. Finally, the predictions of the personalized model can be compared with the experimental outcomes of the specific animal to gain insight on the neural pathways modulating physiological functions and estimating functional outcomes of future stimulation policies.

Validating computational models of the VN with experimentally measured functional responses to neurostimulation in animals is a challenging, yet crucial step, to enable the use of these models for in vivo experiments. Previous models have either lacked validation[24,26] or employed very limited approach that compared experimentally obtained and model-estimated thresholds of different fiber types placed in all fascicles, neglecting the existence of fiber clustering[21]. Due to the absence of the realistic 3D anatomy of the nerve, the distances between active sites and fibers are altered, along with the surrounding tissues, thereby affecting the estimated thresholds. Moreover, because of the lack of precise fiber placement, previous models rely on averaging thresholds across all the fibers in the nerve, reducing the validation to a comparison of ranges with experimentally observed thresholds[26,48]. In contrast, we validated our personalized model using experimentally recorded compound action potentials (CAP) evoked with VNS, active site by active site. The experimental curves of the CAP and corresponding model recruitment curves showed high correlation, demonstrating that the personalized model, with correct order of active sites and thresholds, can be reliably used to understand and model the functional outcomes of experiments. While many C-fibers exist in the nerve and are connected with a range of important functions[68] such as reflex vasodilation triggered by cardioprotective unmyelinated ventricular C-fibers[69], experimentally recorded CAP shows that a very small amount is engaged using cuff electrodes and classical stimulation paradigms[16]. This finding is confirmed by our model estimation. However, the model can guide development of highly spatially selective electrodes toward increased and controlled C-fiber activation.

We compared the $A_{Eff}$ and $B_{Eff}$ fibers activation in the model with experimentally obtained changes in laryngeal EMG and heart rate, respectively. The interplay of their neuromodulation is important for the successful HR variation (modulating $B_{Eff}$), while not inducing coughing or breathing problems (caused by $A_{Eff}$ recruitment). Using VaStim in combination with the experimental data, we discovered that, while all $A_{Eff}$ fibers across the nerve influenced the L-EMG changes, heart rate changes were caused by activation of $B_{Eff}$ fibers located in the specific fascicles. We identified these heart-specific fascicles and estimated the amount of induced HR change based on the simulated neural activity. Our model accurately predicts experimentally recorded physiological changes in laryngeal muscle activation and heart rate. We quantified the therapeutic benefit (change in HR) relative to the side effects (laryngeal activation) both in the experimental results and in the model, revealing that commonly used stimulation paradigms result in significant side effects[9,13,16]. This quantification is crucial, as studies often rely solely on observable but non-measurable side effects[70], or do not precisely state the effects they are considering[23]. Thus, we exploited our model to test whether an alternative stimulation strategy could reduce the unwanted L-EMG while achieving the same target outcome. We found that a specific tripolar configuration allows for a more localized distribution of potential (Fig. 5d upper part), enabling more directed and controlled stimulation than the monopolar one (Fig. 5d lower part). Consequently, with a tripolar policy, the selectivity of stimulation could increase, allowing less unwanted effect and therefore enabling further exploration of therapeutic effects of VNS for dealing with cardiological issues.

Finally, we used VaStim to test the importance of the precise surgical implantation of the helical cuff electrode commonly used in VNS. When prioritizing the enhancement of achievable selectivity, modeling confirmed that the precise placement of the above-mentioned electrode is not crucial, meaning that there is the tolerance in its rotational and translational placement. At the same time, this evidence supports the choice of eight active sites for a cuff electrode, suggesting that these sites adequately cover the circumference of the nerve, regardless of their orientation[16]. On the other hand, the fact that only very poor selectivity can be achieved for non-superficial fascicles, combined with the observed functional clustering, suggests that intraneural electrodes may be required to modulate functions which cluster deeper within the nerve, and indeed has been shown promise for VNS[70]. Moreover, the use of intrafascicular electrodes can improve the selectivity for smaller diameter axons[48], especially if they cluster within fascicles, possibly reducing off-target effects such as laryngeal muscle contractions due to $A_{Eff}$ recruitment. The strong functional implications of spatial clustering indicate that the accurate fiber distribution is crucial to consider when utilizing models to design electrodes and stimulation paradigms, and generally reinforces the importance of spatial selectivity for obtaining therapeutic effects minimizing undesirable side effects.

## Limitations

The anatomical accuracy of the presented model is currently constrained by the distances between the segmented histological slices. Fascicle merging and branching is based on assumptions about the nerve structure. In the human cervical VN fascicle branching have been reported to occur up to every ~560 μm[28]. The impact of considering more densely spaced fascicle branching on simulation predictions remains a subject of future research. Achieving such precision would require a large number of sections which could be obtainable through the use of micro-CT imaging[16] with high spatial resolution. This would also necessitate the automatized segmentation and anatomical reconstruction, which are presently very long and computationally expensive.

The precise fiber placement was performed at the nerve level where immunochemistry was performed, with the fibers extending throughout the length of entire nerve, following its curvilinear trajectory. Consequently, this approach may introduce inaccuracies in the specific positioning of the fibers away from the initial cross-section.

However, as it generally respects the crucial feature of clustering of different fiber types that exist in all VNs, we believe this limitation does not drastically affect the results.

As physical models are always partial representations of reality, a compromise between accuracy and simplicity must be drawn, as too complex models can be unjustifiably costly to develop or simulate, while too simple ones can lead to misleading conclusions. Simplifications include the representation of the surrounding medium as homogeneous saline environment as approximation of the operating conditions, as typically done[17,30,37,44–46,48,49]. Representing precisely the surrounding tissues which would have made recruitment estimations more accurate, but it would have likely not affected the essential conclusions[71]. Moreover, the personalization step we devised should be able to at least partly account for this type of discrepancies. Even higher prediction accuracy may also be obtained by using modality-specific models for afferent and efferent fibers[39].

All methods developed to increase the computational efficacy of the models are applicable for use in different VNS scenarios. However, future simulations involving high-frequency stimulation may require a renewed convergence study to identify time step values that are appropriate for the characterization of faster transients. Also, in case of greatly different configurations (e.g., with intrafascicular electrodes), it may be beneficial to repeat the convergence studies for dynamic discretization (Fig. 2a, b) and longitudinal truncation (Fig. 2c, d).

Moreover, achieving a match between the model estimation of achieved physiological changes and experimentally obtained results would be more precise if we could attain the saturation of the experimental outcome changes. However, estimating saturation thresholds during animal experiments was not feasible due to safety reasons.

### In silico tool to improve selective neuromodulation of the vagus nerve, crucial for expanding its therapeutic use

With continued research and development in this area, VNS could offer even more promising results for patients suffering from various chronic disorders[68]. However, there is a need for more specific and sophisticated techniques to enhance VNS applications, allowing to effectively treat a broader range of medical conditions. Precise neuromodulation involves advancements in technology and approaches that enable us to target specific nerve fibers while avoiding the stimulation of the rest of the nerve. Selectivity of VNS is still a crucial unsolved challenge.

Commonly used VNS paradigms often induce unbearably strong laryngeal muscle contractions which obstructs the airways and induces throat pain with intense coughing. With our model, we were able to confirm the experimentally obtained results indicating that the stimulation with the cuff electrode and commonly used paradigms has very limited selectivity while induces heart rate changes[16,72]. We are suggesting that these side effects could be reduced with a novel stimulation paradigm that could be tested in the model. One of the approaches can be the use of multipolar stimulation[23,72]. Moreover, VaStim can be used as a platform to design and optimize electrode designs, as explored in previous studies[24,48,49,71], enriched by the accurate representation of clustering of fiber types and their functionalization through physiological experiments.

Presented modeling effort can be a valuable tool for testing the stimulation policies for increasing the stimulation selectivity. Since pigs' cervical vagus nerves have similar diameter as the human nerve, albeit with much higher number of fascicles[31,42], these stimulation paradigms could also be beneficial for the human applications. Our complete realistic model is a free and publicly available tool, accompanied by a web-based platform, that can be exploited to help the researchers to optimize the VNS by simulating novel paradigms or electrode designs. Together with testing optimal ways to increase the

fascicular selectivity, thanks to its great accuracy in representing the anatomical organization of fibers across the nerve, it can work toward fiber-targeted stimulation. That is even more valuable in human VNS as lower number of fascicles indicates higher mixture of physiological outcome-specific fibers in the single fascicle[16]. Following the in silico results, pig animal models can be used for testing the proposed paradigms, before translating these approaches to humans. Our approach can be easily used with comparable anatomical data collected through recently initiated studies in human vagus anatomy[73]. Finally, this framework, combining latest experimental VNS discoveries with computational approaches, not only expands the scientific understanding of vagus neural mechanism, but also hold significant promise in advancing the next generation of safe and effective neuromodulation therapies and devices.

## Methods
### Ethical statement
All animal protocols and surgical procedures were reviewed and approved by the Animal Care and Use Committee of Feinstein Institutes for Medical Research and New York Medical College.

### Immunohistochemistry of nerve samples
After the humane euthanasia of two pigs (male Yucatan ~40 kg, ~1 year old, Premier BioSource) via Euthasol injection, a meticulous surgical dissection procedure was undertaken, exposing and isolating both the left and right cervical vagus nerves, from above the nodose ganglion down to the termination point in the thoracic vagus. To enable subsequent imaging and tracking of vagal branches, sutures were delicately placed using a 6-0 fine suture loop, affixing them to the epineurium of each branch near its emergence from the trunk. Detailed photographs were captured before and after the nerve extraction process, providing comprehensive documentation of the nerve trunk and its associated labels. Following this, the isolated nerve samples underwent meticulous fixation in 10% formalin for a duration of 24 h. For the immunohistochemistry (IHC) studies, both the left and right vagus nerves ($n = 2$) were meticulously sectioned at a thickness of 5 microns at four different levels: initially at the nodose ganglion, followed by the upper cervical level, mid cervical level, and lastly, the lower cervical level. From each of these segments, a total of 200 sections were obtained, with each section measuring 5 μm in thickness. Approximately 25 of these sections were then selected for the IHC study. These sections underwent staining for specific markers, including myelin basic protein (MBP), neurofilament (NF), choline acetyltransferase (ChAT), tyrosine hydroxylase (TH), and voltage-gated sodium channels 1.8 (Nav1.8). Additionally, a distinct set of adjacent sections was utilized for H&E staining to identify cellular structures such as endoneurium, perineurium, and epineurium. These staining procedures strictly adhered to standard IHC protocols, encompassing essential steps such as antigen retrieval, blocking, incubation with primary and secondary antibodies, and rigorous washing steps. The imaging of the stained sections was conducted utilizing advanced equipment, including the ZEISS LSM 900 confocal laser scanning microscope and the BZ-X800 all-in-one fluorescence microscope.

### Segmentation and characterization of single fibers from IHC images
The vagus nerve is segmented using a combination of standard computer vision and deep learning techniques from immunohistochemical images[16,74]. We employed a systematic approach to detect specific anatomical markers in the nerve: neurofilament, myelin, ChAT, and Nav1.8. For each marker, a brightness threshold was empirically selected to classify positive pixels. Blobs or connected components of these pixels were then identified, with noise below the said threshold discarded. For detailed fiber segmentation, the deep learning model

Mask R–CNN was initially used to detect axons. It was then complemented with traditional methods to address overlapping neurofilament blobs that the deep learning model missed, achieving a fiber detection accuracy of 96%.

After segmenting neurofilaments and assigning myelin to detected fibers, various features, such as fiber diameter, perimeter, and myelin thickness, were extracted. Fascicles were manually annotated to extract features at that level. To generate a comprehensive dataset, manual annotations of Nav1.8-stained images were made to identify the density of C-fibers within Remak bundles, enabling the creation of a linear equation to estimate fiber counts in other bundles. Image pre-processing, like adaptive histogram equalization and color-clipping, was vital to enhance contrast and eliminate noise. Finally, fibers were classified into four distinct categories based on morphology and function: myelinated and unmyelinated efferent and afferent.

## Creating a realistic 3D model of the cervical vagus nerve

Four distinct histological slices (C1 to C4) of the explanted vagus nerve were extracted from the mid-cervical regions of both animals (labeled M1 and M2). C1 was located at 40 mm from the nodose ganglion and served as a reference for the other slices. Toward the thoracic region, these were located at distances of 10 mm (C2), 25 mm (C3) and 35 cm (C4) from C1, spanning a total length of 35 mm.

We manually segmented the contours of the epi- and endoneuria of the histological slices using ImageJ with the NeuronJ plugin[75,76], yielding the anchor point coordinates of two-dimensional splines. These were imported into Solidworks, respecting the correct distances between the individual slices. All splines were then simplified by applying Solidworks' *Simplify Spline* function with a tolerance under 0.005 mm, until no further reduction in spline points could be achieved. This did not appreciably alter the shape of any contour, but significantly improved the rendering efficiency and later simplified the mesh. Next, the fascicular contours at the different cross sections were matched visually to determine the branching and merging paths of each fascicle, which were then reconstructed using the *Lofted Boss/Base* feature. Using the corresponding cross sections, the epineurium was also lofted. Lastly the feature *Interference Detection* was utilized to detect any intersections between fascicles as well as fascicles protruding outside of the epineurium, which were manually resolved by adding appropriate guide splines to the lofts.

The helical cuff electrode used in the experimental setting was modeled after the manufacturer specification. The electrode is composed of 8 surface active sites radially distributed around the nerve. Each active site is squared and has an area of 0.25 mm². They are laterally separated by 0.5 mm. The electrode substrate was wrapped around the epineurium using the *Wrap* function.

## Setting up a finite element model for vagus nerve stimulation

In order to solve the electrical fields that the active sites induce during stimulation, the 3D model was imported into COMSOL Multiphysics. Here a homogeneous saline cylinder with radius of 35 mm was added to emulate the interoperative environment, with the diameter chosen by a convergence study so that the ground-at-infinity condition was sufficiently approximated[30,37,44,45]. The perineurium was modeled by adding a thin layer contact impedance to each fascicle boundary, with the surface thickness being set to 3% of the average area over all four cross sections of the corresponding fascicle. This approach leaned on the common assumption that the endoneurium thickness equals about 3% of the fascicle area[37,44,46,48,77]. Epineurium, perineurium, electrode substrate and saline were assumed to be isotropic, whereas the endoneurium was modeled with anisotropic conductivity, due to the presence of fibers within. The precise conductivity values are based on previous modeling studies[44], and listed in Supplementary Table 2. To ensure that the anisotropy is correctly oriented with respect to the fascicular cross sections, and not the global coordinate system, we

used COMSOL's *Curvilinear Coordinates* feature. In particular, the anisotropic vector field was aligned to the streamlines determined by a diffusion study, in which the fascicle hulls were defined as walls and their flat ends as inlets and outlets. Both nerve models were meshed resulting in about 15 to 20 million tetrahedral elements. For low frequency stimulation, it is adequate to exploit the quasi-static approximation of Maxwell's equations in order to calculate the electric fields Vext[78] (Eq. (1)), where σ is the conductivity:

$$\nabla \cdot \sigma \nabla \mathbf{V_{ext}} = 0 \tag{1}$$

As this equation is linear, it is possible to determine the electric fields resulting from an arbitrary multipolar stimulation by scaling and superimposing electric fields which were independently determined for all active sites under consideration of unit charge injection[44] from surface current sources corresponding to each active site.

## Populating the model with histologically accurate fiber data

For both nerve models, precise fiber data was obtained from the immunohistological analysis of cross section C2 in a fascicle-wise fashion. Fiber types and modalities were differentiated by correlating the responses of NF (neurofilament), MBP (myelin basic protein), ChAT (choline-acetyl-transferase) and Nav1.8 stains[16]. We extracted precise locations and diameters of all myelinated fibers. Due to imaging resolution limits, single unmyelinated fibers could not be labeled. Instead, we identified centroids and areas of clusters of unmyelinated fibers. A supplementary study with higher resolution was conducted, yielding the empirical diameter distributions for both afferent and efferent unmyelinated fibers as well as a linear correlation between the cluster area and count of contained fibers[16]. Using these results, the fiber count for each cluster was estimated and the diameters were assigned by inverse sampling gamma distributions (Eq. (2)) which had been fit to the empirical diameter distributions (Table 1). The precise locations of unmyelinated fibers were generated by uniformly sampling coordinates within a circle centered on the centroid of the cluster, with area equivalent to that of the cluster:

$$p(x) = \frac{1}{b^a \Gamma(a)} x^{a-1} e^{-\frac{x}{b}} \tag{2}$$

The fibers were now integrated into the model by performing fascicle-wise matching between centroids of the modeled fascicles (of cross section C2) and the centroids of all corresponding fibers. Fibers placed outside of the endoneuria were subsequently removed.

To reduce the computational costs we subsampled the simulated fibers. A convergence study was conducted to obtain the minimum number of fibers necessary to accurately reproduce the recruitment curves of the full data set. Using the criterion that the mean absolute deviation of recruitment curves should be lower than 5% for each fiber type, we determined that it is suitable to simulate at least 10'000 fibers per nerve (Supplementary Fig. 1). During subsampling, care was taken to ensure that the proportion of the individual fiber types as well as their modalities across the whole data set does not change substantially. The exception was to always sample all Aα fibers, due to their low occurrence, and to remove all myelinated fibers with a diameter smaller than 1 μm or larger than 18 μm due to the expected inaccuracy of the extrapolated MRG model for these values.

The fibers were propagated along the nerve based on their placement in the initial cross section as well as the curvilinear coordinates extracted from COMSOL along each fascicle. This further ensured the precise alignment of each axon to the anisotropic vector fields of the endoneurium. Moreover, it ensured smooth fiber trajectories along fascicular branching. Table 2 lists the number of fibers placed and

**Table 1 | Parameters of the gamma distributions fitted to empirical diameter distributions of unmyelinated afferents and efferents**

| Modality | a [–] | b [–] | Interval [µm] |
|---|---|---|---|
| Afferent | 80.8746 | 0.00456138 | [0.3,0.45] |
| Efferent | 8.54092 | 0.227013 | [0.73,4.36] |

**Table 2 | Population sizes of the different fiber types for both nerve models**

| Fiber type | M1 | | | M2 | | |
|---|---|---|---|---|---|---|
| | Histology | Sampled | Fraction [%] | Histology | Sampled | Fraction [%] |
| Aα | 8 | 7 | 0.069 | 14 | 7 | 0.069 |
| Aβ | 1361 | 57 | 0.564 | 1441 | 27 | 0.265 |
| Aγ | 3212 | 138 | 1.365 | 3646 | 498 | 0.813 |
| Aδ | 18,923 | 746 | 7.381 | 22,905 | 83 | 4.879 |
| B | 22,537 | 942 | 9.320 | 22,400 | 513 | 5.026 |
| C | 215,905 | 8217 | 81.3 | 429,942 | 9080 | 88.95 |
| Total | 261,946 | 10,107 | 100 | 480,348 | 10,208 | 100 |

Reported are the numbers of the full immunohistological analyses and subsampled data sets, with the latter only considering fibers that were also successfully propagated along the models. Note that the full data set also includes fascicles which were not incorporated into the models. The relative proportion of the fiber types is reported with respect to the "sampled" column.

successfully propagated along both nerve models in the context of this paper.

### Extracting extracellular potentials along fibers

Once the precise fiber coordinates in 3D space are known, it is possible to extract the extracellular potentials along the fibers resulting from unit charge injection by an active site. For this, the set up COMSOL model is solved for each active site individually by applying a reference current density of $1 \, A/m^2$. To obtain the extracellular potential corresponding to a 1 µA charge injection, this solution is then interpolated at discrete points along the fibers and scaled by the electrode area. The potential was extracted at the center of each section of each fiber.

### Modeling the membrane dynamics of myelinated and unmyelinated fibers

To simulate the response of fibers to extracellular stimulation, or, more precisely, the time-course of their membrane potential, it is necessary to model the ion channels as well as passive components. The dynamics of ion channels are usually formulated as modified nonlinear Hodgkin–Huxley equations. It must further be considered whether a given section is myelinated or not, as the myelin sheath acts as a local insulator. The equivalent electrical circuits for sections of myelinated and unmyelinated fibers are illustrated in Supplementary Fig. 8.

Based on a comprehensive review[33], where several models of unmyelinated fibers were compared, the Tigerholm (TH) model was selected to model unmyelinated fibers[36]. It was deemed to reproduce data from single-fiber recordings most accurately, albeit the experimental recovery cycle could not be captured. Furthermore, the TH model has been incorporated into modeling pipelines similar to the one presented in this work[26]. The TH model was implemented using the previously published files[33]. However, a discrepancy between the corresponding conductance values of the pump and $K_{Na}$ channels and those listed in the original publication[36] was observed. The values of the latter were used (see Supplementary Table 3), which we confirmed to accurately reproduce the action potential current traces[36].

For myelinated fibers the frequently used McIntyre-Richardson-Grill (MRG) model was chosen[17,26,37,38,44,45,47,50,58,79,80]. It compartmentalizes

the region between two nodes of Ranvier into ten distinct sections, the parameters of which are diameter dependent. As the original parameters were presented for only a small set of discrete fiber diameters, it is necessary to inter- and extrapolate their values such that the model can be applied to fibers of arbitrary diameters. In this context polynomial interpolation has previously been performed[26], which was also adopted in this project although with slightly deviating results (see Supplementary Fig. 9 and Supplementary Table 4). Furthermore, a rounding imprecision of the published leakage conductance caused a small transient membrane hyperpolarization ($V_{init} - E_{leak}$) to occur when the fiber was completely at rest. This was resolved by recalculating the leakage conductance $g_{leak}$ at every node such that the leakage current exactly balances all ionic currents $i_{Na}$, $i_{Nap}$, and $i_K$ at rest (Eq. (3)), an idea which was also implemented for the previously published TH model implementation[33]. This causes the leakage conductance to only deviate slightly from its original value:

$$g_{leak} = -\frac{i_{Na} + i_{Nap} + i_K}{V_{init} - E_{leak}} \quad (3)$$

All NEURON functionality was implemented using the NEURON-Python interface. For both the TH and MRG models, the original HOC scripts were ported to Python while the model description (NMODL) files were not altered.

### Determining the recruitment threshold of a fiber

To execute a simulation in NEURON the stimulation paradigm must be encoded in a pulse matrix of the form which contains the injected current of each active site in µA along with the time stamp at which a change in stimulation amplitude of any active site occurs. After a simulation is finished the current process is cleared of all remaining NEURON objects by iteratively deleting all detectable distributed membrane mechanisms, point processes as well as sections. This does however not remove the *extracellular* mechanism, which fundamentally alters the structure of the underlying Jacobian matrix to represent the additional layers in the cable model. In some cases, this might affect a subsequent simulation, which is circumvented by manually resetting the extracellular potential of each node to 0.

The method outlined above is utilized as part of a bisection algorithm to determine the recruitment threshold of a fiber, which indicates by what factor the charge injection amplitudes in the pulse matrix must be scaled to elicit an action potential. Our pipeline natively supports execution of this algorithm on a High Performance Cluster (HPC), including automated bidirectional file transfer.

### Dynamic discretization of unmyelinated fibers

We spatially discretized unmyelinated fibers using sections of varying length, to optimally trade-off the number of sections and simulation accuracy. In this regard, we developed an algorithm (see Supplementary Listing 1) which assigns section lengths based on local estimates which evaluate how relevant a fiber region is for accurately predicting the physiological response of the whole fiber. This idea is motivated by the fundamental result that there is an approximately linear correlation between the subthreshold membrane (de)polarization $V_{mem}$ at node $i$ and the value of the second spatial derivative of the extracellular potential $V_{ext}$ at node $i$ along the axon (known as activation function, Eq. 4)[55], where $\rho_a$ is the axoplasm resistivity and $c_m$ the membrane capacitance:

$$\frac{\partial V_{mem,i}}{\partial t} \approx \frac{d}{4\rho_a c_m} \frac{\partial^2 V_{ext,i}}{\partial z^2} \quad (4)$$

A prerequisite for the dynamic discretization algorithm is that the extracellular potential of each fiber has been extracted from COMSOL by interpolating the solution of the electric field at fine resolution. For

our nerve models with a total length of 35 mm a resolution of 10 μm was selected, which yielded about 3500 nodes per extracellular potential per fiber. The dynamic discretization algorithm then proceeds by determining an estimate of the second spatial derivative of the extracellular potential along each fiber. Here, the extracellular potential is first padded by linear extrapolation at both ends of the fiber by 5% of its total length. The 2nd order central differences are now evaluated, which yields a noisy estimate of the second spatial derivative. Smoothing is subsequently performed using a 2nd order forward-backward Butterworth low pass second-order sections (SOS) filter. The cutoff frequency of this filter was set to 0.001. After smoothing the padding is removed. The positive and negative parts of the estimated second derivative are then normalized independently, such that all values fall within the range [0,1]. This renders the estimate invariant to the scale as well as sign of the applied extracellular potential. The fiber is then discretized so that each section length is proportional to the local second derivative within a predefined minimum and maximum section length. Lastly, the extracellular potential is resampled at the newly determined fiber nodes.

Pseudocode of the proposed dynamic discretization algorithm for a fixed extracellular potential is given in Supplementary Listing 1. Note that generalization to multipolar stimulation with arbitrary stimulation paradigms and spatial distribution of electrodes is straightforward.

### Longitudinal truncation of fibers
To determine the truncation threshold for the normalized extracellular potential, based on which the truncation locations are defined, a study is conducted a priori on a sample set of fibers. This study is performed independently for both myelinated and unmyelinated fibers, and the result is then applied to all fibers in the nerve model.

For the study, fibers with discretely assigned diameter values are randomly placed across the nerve. The initial recruitment threshold of each fiber is then evaluated using the full-length fiber. Next, a bisection algorithm along the normalized extracellular potential is executed to determine an appropriate value for the truncation threshold, whereas the fibers are truncated to the left and right from the outermost nodes corresponding to this value of the truncation threshold. If the recruitment threshold determined for the truncated fiber deviates by less than 0.5% from the full-length recruitment threshold, the truncation threshold is increased to further truncate the fiber; otherwise it is decreased, elongating the fiber again. The termination condition is that, for two consecutive iterations, the truncation threshold determined by the bisection algorithm corresponds to the same nodes, for which the recruitment threshold deviation is furthermore less than 0.5%. The determined truncation thresholds are grouped by fiber diameters. For both myelinated and unmyelinated fibers, a spline is then fit through the 0.5th percentile of all diameters to obtain a conservative diameter-dependent estimate of the threshold ratio. Pseudocode of the proposed longitudinal truncation algorithm for a fixed extracellular potential is given in Supplementary Listing 2.

### Increasing the integration time step
The sweep over different time step values was conducted by evaluating all fibers in M1 under application of a cathodic pulse to active site 1, with a total simulation time of 3 ms and a threshold precision of 0.01 for MRG fibers and 0.1 for TH fibers.

### Studies for the improvement of computational efficiency
The results in Fig. 2 and Supplementary Fig. 3 were obtained by stimulating M1 using the following parameters for the baseline model and presented methods (if applicable):

### Dynamic discretization
The minimum and maximum section lengths were set to 20 μm and 200 μm, respectively. We discretized the fixed-length baseline models with sections of 20 μm length (yielding ~1750 sections for ~35 mm long fibers).

### Longitudinal truncation
A study with a threshold accuracy of 0.01 for MRG fibers and 0.1 for TH fibers was conducted on an artificial data set. Fiber diameters were sampled from the discrete sets {1, 1.5, ..., 18} μm (myelinated fibers) and {0.3, 0.8, ..., 4.8} μm (unmyelinated fibers). Each diameter was assigned 50 times, but due to imperfect fiber propagation only 1694 myelinated and 478 unmyelinated fibers were eventually placed in the model.

### Time step increase
We increased the time step from 5 μs in the baseline model to 13 μs in the modified model.

We set the total simulation time to 3 ms and used a recruitment threshold accuracy of 0.1 μA. In Fig. 2 the nerve was stimulated using a monopolar cathodic pulse with 500 μs pulse width, whereas in Supplementary Fig. 3 the nerve was stimulated using a monopolar anodic pulse with 500 μs pulse width. Relative deviations of recruitment thresholds due to incorporating the three presented methods were evaluated as in Eq. (5):

$$\delta_{\lambda,mod} = \frac{\lambda_{mod} - \lambda_{orig}}{\lambda_{orig}} \tag{5}$$

where $\delta_{\lambda,mod}$ corresponds to the relative deviation, $\lambda_{mod}$ to the recruitment threshold obtained with the optimized method, and $\lambda_{orig}$ to the recruitment threshold obtained with original method. Summary of the improvement in computational time is given in Supplementary Tables 1 and 5.

### Comparison of models with realistic and linearly extruded fascicle branching
Models with linearly extruded fascicles were created using the cross sections in which fibers are initially placed for both nerve models. This ensures that the same fibers are present in corresponding realistic and extruded models. The electrode was placed on three different locations, at a longitudinal distance of 2.5 mm, 7.5 mm and 12.5 mm from the extrusion cross section in direction of the thoracic region. For each fiber in each model, recruitment thresholds were estimated by independently applying a cathodic pulse to each active site. The relative deviation of the recruitment thresholds of corresponding extruded and realistic models ($\delta_{\lambda,extr}$) was evaluated as in Eq. (6):

$$\delta_{\lambda,extr} = \frac{\lambda_{extr} - \lambda_{real}}{\lambda_{real}} \tag{6}$$

where $\delta_{\lambda,extr}$ corresponds to the relative deviation, $\lambda_{extr}$ to the recruitment threshold obtained with the extruded model, and $\lambda_{real}$ to the recruitment threshold obtained with realistic model. The means of the distributions were tested by a one-sample, two-tailed t-test.

### Precise fiber placement can improve selectivity of synthesized stimulation policies
To create models with uniformly distributed fibers, the locations of all fibers placed in the histologically accurate models were reassigned using Monte-Carlo rejection sampling. The magnitude of clustering of each fiber group was evaluated using silhouette values, which quantify clustering as the difference between the cohesion within the cluster and the separation with other clusters, and ranges from −1 and +1[56]. Curves of the spatial selectivity indices $\eta_{group}$ were evaluated using Eq. (7):

$$\eta_{group} = \mu_{group} - \mu_{\neg group} \tag{7}$$

as a cathodic pulse stimulation policy was scaled, where $\mu_{group}$ is the recruited fraction of fibers belonging to the target group and $\mu_{\neg group}$ is the recruited fraction of non-target fibers. Such metric is commonly used[49,67] and is easily interpretable. As other spatial selectivity indexes (as in Eq. (8)), it ranges from −1 (recruitment of all non-target fibers, no recruitment of target fibers), to 1 (recruitment of all target fibers, no recruitment of non-target fibers). For each group, we selected the best active site as the one with spatial selectivity curve with the largest mean value. The presented recruitment curves were also obtained for the same active sites. The functional consequences of fiber clustering were studied by comparing the models with histologically accurate fiber location with four models where the same fiber population was spatially shuffled across the nerve (Fig. 3b and Supplementary Fig. 5).

## Physiological experiments on new subjects

After sedation with Telazol (2–4 mg/kg), three pigs (male Yucatan ~40 kg, ~1 year old, Premier BioSource) were placed on a table in a supine position with body temperature maintained between 38 °C and 39 °C using a heated blanket. Anesthesia was induced with Propofol (4–6 mg/kg, i.v.), and maintained with isoflurane (1.5-3%, ventilation). Then, a 4–5 cm long incision was cut on the cervical region of the neck. After dissecting the underlying tissues, the vagus nerve was identified and isolated. One or two helical cuff electrodes (CorTec, Germany) were placed on the rostral (around 2 cm to nodose) and/or caudal sites with a distance of 4 to 5 cm. The helical cuff included 8 contacts (0.5 mm by 0.5 mm) evenly distributed (1 mm between two contacts) around the circumference and 2 ring-shaped (0.5 mm by 7.4 mm) return contacts. EMG electrodes, which were teflon-insulated stainless-steel wires with de-insulation of 1 mm at the tip, were inserted in the thyroarytenoid (TA) of laryngeal muscle with a needle after blunt dissection of the hyoid bone. All procedures were performed using sterile techniques and approved by the animal care and use committee of Feinstein Institutes for Medical Research and New York Medical College.

ECG signals were recorded and amplified (FE238, ADI) by using a 3-lead patch electrode configuration. Blood pressure was recorded from the aorta with catheter (SPR-751, Millar Inc) after amplification (FE228, ADI). All physiological signals were digitized and acquired at 1 kHz (PowerLab 16/35, ADI). Neural signal and EMGs were sent to the data acquisition system through a headstage (RHS 32, Intan Tech). It was filtered (60-Hz notch), amplified, and digitized at a sampling frequency of 30 kHz with an acquisition system (Intan 128 recording/ stimulating, Intan Tech).

Biphasic stimulus pulses were delivered using an isolated constant current stimulator (STG4008, Multichannel Systems) through each of the 8 contacts of the cuff electrode. Physiological threshold (PT) was detected from stimulus trains of 10 s duration (30 Hz, pulse width 200 μs). There were at least 30 s long waits between successive trains to ensure that physiological measurements had reached a steady state before a new train was delivered.

After removal of the direct current (DC) component, the evoked compound action potentials (eCAP) were averaged from each stimulus and classified into different fibers groups based on the conduction speed. The first peak (0.8 ms) shows A-fiber response (conduction speed, 5–120 m/s), and the late peak (-1.5 ms, ~3 ms) shows A-delta/B-fibers (conduction speed, 5–15 m/s). Recruitment maps of different fiber responses (peak-to-peak amplitude) in the recording cuff (8 contacts) were derived as second cuff was stimulated (8 contacts).

## Model personalization to new experimental subjects and validation

We developed a method to adapt these highly detailed computational models to new experimental subjects for which no histology can be obtained (in vivo). We first simulated the stimulation paradigm used in the experiment (monopolar, biphasic, charge-balanced squared pulse of pulse width 200 μs). We applied a linear regression model with no

intercept with input the model recruitment curves of Aα efferents per active site and output the experimental recruitment curves for laryngeal EMG per active site. We then found the scaling factor that applied to the fiber thresholds predicted by the model maximized the R2 of the linear regression model. Since the electrode is radially symmetric, we computed this maximized R2 for all possible shifts of active sites around the nerve circumference ($n = 7$), and chose the shift of active sites that, together with the scaling factor, predicted best the laryngeal EMG (Fig. 4a, b). We computed the personalization steps for each model (M1 and M2) and experimental subject (S1, S2, and S3) for a total of 6 personalized pairs of models-experimental subject. For each pair, we then compared the final R2 with the R2 of the original non-personalized model with a Wilcoxon signed rank test. We then validated the personalized model by comparing the model recruitment curves for Aα and Aβ efferents with the corresponding fast compound action potential recorded experimentally[16], per active site, normalized to [0,1] between minimum and maximum CAP recorded per-subject. We then computed Pearson's correlation coefficient between recruitment curves (Fig. 4c ii and Supplementary Fig. 6), and we compared the thresholds to obtain 10% to 70% recruitment in model and experiment (in steps of 10%), using a Wilcoxon signed rank test (Fig. 4c iii and Supplementary Fig. 6). The higher boundary of 70% was chosen since Pig 3 did not reach higher recruitment values in the experimental range. Finally, we compared the ranking of active sites between model and experiment by computing the Spearman's correlation coefficient between the thresholds predicted by the model and by the experiment per active site for each recruitment level (10% to 70%). Finally, to exclude that the result was a result of chance, we performed the same computation of correlation randomly shuffling the active sites in the model 100 times. We compared the two distributions of correlation coefficients with a Wilcoxon rank sum test (Fig. 4c iv and Supplementary Fig. 6).

## Prediction of hearth rate variation and model-guided reduction of off-target effects

We predicted the experimentally measured heart rate reduction with the fiber activation estimated by personalized models through multiple linear regression. The model inputs the number of recruited $B_{Eff}$ in each fascicle to the predictor (for a certain active site and injected current level), which outputs the predicted heart rate reduction. No intercept was included since there must be no heart rate variation when no fiber is activated. The weights were constrained to be positive to reflect the hypothesis that $B_{Eff}$ activation causes heart rate reduction (with positive absolute values), and not increase. The fascicles containing a low number of $B_{Eff}$ were excluded using k-means clustering with $k = 2$. Further, we excluded the fascicles where no $B_{Eff}$ was recruited within the current range of the experiments (Fig. 5a, b and Supplementary Fig. 7). We then computed the RMSE between model prediction and experiment (Fig. 5c and Supplementary Fig. 7). From the multiple linear regression we extracted the combination of fascicles responsible for experimental HR changes (weight > 0). Using selected fascicles, we identified for each personalized model the active site with the lowest threshold to obtain 25% recruitment of $B_{Eff}$ across the target fascicles. In total, 25% was defined by selecting the experimental charge for targeted HR change (2500 μA[16]) and by computing and averaging the $B_{Eff}$ recruitment for each personalized model. We then computed for that active site the recruitment curves for Aα across the whole nerve (correlated with L-EMG activity) and for $B_{Eff}$ in the fascicles which were assigned a non-zero weight by the linear regression. Then, we performed an additional study, simulating tripolar stimulation with a charge-balanced asymmetric waveform. The active site identified in the previous step was set as pseudo-cathode, with a first cathodic pulse with 200 μs width and a second balancing anodic pulse with 20 μs width and one tenth of the amplitude. The active sites on either side of the pseudo-cathode were configured as pseudo-anodes,

with opposite sign and each half amplitude than the pseudo-cathode (Fig. 5a ii, d). We computed the recruitment curves for Aα and target $B_{Eff}$ as for the monopolar case (Fig. 5d). We then identified the level of recruitment of Aα that corresponded with the 25% recruitment of target $B_{Eff}$ for each personalized model and compared the distributions between the monopolar (experimental) and the tripolar case with a Wilcoxon signed rank test (Fig. 5f). Finally, we compared the current thresholds required to obtain the 25% recruitment of target $B_{Eff}$ for both stimulation paradigms and compared them with Wilcoxon signed rank test (Fig. 5g).

### Evaluation of electrode performance due to surgical placement uncertainty

For each nerve, eighteen models with different electrode placements were created. The three locations were chosen such that the center of gravity of all active sites coincides with a point at a longitudinal distance of −5 mm, 0 mm, or +5 mm from the center of the nerve model. At each location the electrode was rotated by the angles 0°, 60°, 120°, 180°, 240° and 300°. Using a cathodic pulse the recruitment thresholds of all fibers were generated for all active sites of all models. The best spatial selectivity indices of all fascicles were then calculated using Eq. (8):

$$\eta_i = \max_{\{AS1,AS2,\ldots\}} \left\{ \mu_i - \frac{1}{m-1} \sum_{j=1, j\neq i}^{m} \mu_j \right\} \qquad (8)$$

where $\mu_i$ is the fraction of recruited fibers in fascicle $i$ and $m$ is the total number of fascicles[44]. All fascicles with $\eta < 0.7$ were omitted from further analysis. To investigate the effect of electrode rotation, the selectivity indices of all six electrode angles were grouped for each of the three longitudinal locations; to analyze the effect of electrode translation, the selectivity indices of all three longitudinal locations were grouped for the six electrode angles; and to analyze the combined effect of electrode rotation and translation, the selectivity indices of all eighteen models were grouped. For each fascicle, the range spanned by $\eta$ over all values in each group was determined. These ranges of $\eta$ were plotted in Fig. 6d–f.

### Online platform

The platform uses *Three.js* for in-browser 3D rendering with *WebGL*. Since it is written in client-side *JavaScript*, it is fully static and can therefore be hosted on any web server. It requires the offline pipeline to export the components of the model in formats which can be read natively by *JavaScript*. Geometries of nerve and electrode are saved in STL format, which can be natively rendered by *Three.js*; numeric data (i.e., fiber trajectories and thresholds) are exported into a plain binary format together with metadata (number of dimensions and size of each dimension), allowing the reading and writing of multidimensional arrays; and model specifications are saved as *JSON* files.

### Technical specifications

The current iteration of the pipeline employs MATLAB 2022a, Python 3.9.12, COMSOL Multiphysics 5.6, and NEURON 8.2.1. Histological slices were segmented with ImageJ 1.52p and NeuronJ 1.4.3. The 3D models were created using Solidworks 2020 SP04. NEURON simulations were performed on the Euler cluster, operated by the High Performance Computing group at ETH Zürich, employing 48 cores of the AMD EPYC 7H12 CPU with 2.6 GHz nominal and 3.3 GHz peak clock speed as well as 98304 MB RAM with 3200 MHz clock speed.

### Statistical analysis

All statistical analyses were performed on MATLAB R2020a (The MathWorks, Natick, USA). All data were reported as mean values ± SD (unless elsewise indicated). The normality of data distributions was evaluated with a one-sample Kolmogorov–Smirnov test. Significance

levels were 0.05 unless differently specified. Asterisks in figures indicate the level of statistical significance, $*p \leq 0.05$, $**p \leq 0.01$, $***p \leq 0.001$.

### Reporting summary

Further information on research design is available in the Nature Portfolio Reporting Summary linked to this article.

## Data availability

All data supporting the findings of this study are available within the article and its Supplementary files. Any additional requests for information can be directed to, and will be fulfilled by, the corresponding authors. The data required for the visualization of results on the online platform was deposited on Zenodo[64] and publicly accessible at https://zenodo.org/records/11219384. Source data are provided with this paper.

## Code availability

All original code, including modeling pipeline and online platform, was deposited on Zenodo[64] and publicly accessible at https://zenodo.org/records/11219384.

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

## Acknowledgements

We thank Felix Camagay, Thomas Sudan, and Pablo Benlloch Garcia for their contribution to the initial model development. The funders had no role in the experimental design, analysis, or manuscript preparation or submission. All authors had complete access to data. All authors authorized submission of the manuscript, but the final submission decision was made by the corresponding authors. This project has received funding from the European Research Council under the European Union's Horizon 2020 research and innovation program (no. 759998 "FeelAgain" to S.R.), from the Swiss National Science Foundation (no. 197271 "MOVEIT" to S.R.), and from the project IDEJE by Science Fund of the Republic of Serbia (DiabeticReTrust no. 7753949 to N.K.S.). The work has been partially supported by a grant by United Therapeutics Corp. to S.Z.

## Author contributions

S.R. and S.Z. conceived the study. F.C. led the development of codebase and models. A.C. supported the models development. R.J. implemented the computational optimizations and obtained their results. F.C. and R.J. performed the studies for evaluating the realistic nerve structure and electrode positioning effect. F.C., N.K.S., and N.G. designed model personalization, validation, and application to optimize stimulation paradigms. F.C. implemented the algorithms and obtained the results. T.Z. conceived and coordinated the fiber data extraction. N.J. performed immunohistochemistry staining and histological analysis, W.S. performed the surgery, W.S. and S.Z. designed and performed the physiological experiments, and collected the experimental data. F.C., R.J., N.K.S., N.G., and A.C. produced the figures. V.T. performed the experiments and extracted the fiber data. N.K.S. coordinated the work. S.R. supervised and guided the analysis. F.C., N.K.S., R.J., N.G., A.C., and S.R. wrote the manuscript. All authors edited and proofread the manuscript.

## Competing interests

The authors declare no competing interests.
