## [Peer Review File · Nature Communications]

REVIEWER COMMENTS

Reviewer #1 (Remarks to the Author):

This contribution describes the development of realistic *in silico* models for vagus nerve stimulation. These models have realistic geometries and are optimized both to reproduce with high fidelity *in vivo* experimental observation and reduce the overall computational cost. Authors prove the viability of their model by proposing and validating a novel tri-polar stimulation strategy to control heart rate using VNS without reduced un-wanted motor or muscular activation. To do so, authors also include models for C-fibers which should be considered in VNS, adding to the novelty of the model. The overall work and results are impressive, of good quality and answer fundamental questions and challenge in the field of neuromodulation, and more specifically in field of vagus nerve stimulation. These elements justify in my opinion to accept this contribution for highly selective publication; however, authors should improve the overall completeness of the document on one major point and several minor points further detailed bellow.

Major modification:

Authors state on lines 538-539: “We have developed a novel method to personalize the models for a new animal undergoing experimental VNS testing”. This sentence has a strong meaning, the word ‘undergoing’ meaning that *in silico* parameters exploration can be performed during experiments or surgery.

Authors presents some elements that clearly support this element, such as computational cost reduction (running the model is in the range of minutes instead of tens of hours). However, the overall experimental timeline is not clear and not sufficiently described and elicited. This setup is partially described in fig. 4.a, however one point is not clear and is in contradiction with the personalization (or the use of the word personalization by the authors): modeling part begins with a realistic model of cervical VN, however, the geometry is constructed from immunohistochemistry of nerve sample (described in method) that supposes the euthanasia of the subject. It is unclear if the realistic geometry is subject dependent, and in this case, models are not usable during undergoing experimental VNS testing, or if geometrical consideration are not subject specific and then the personalization has to be genuinely better described by authors.

This should be clearly described in the text and supported by a figure describing the experimental path or timeline with the different *in vivo/in silico* phases, perhaps as an introduction to the method section.

Minor modifications:

- Authors refer in the text to two animal models that are named M1 and M2 (line 136), however they are renamed A1924 and A6050 (Fig. 3 for instance but not only). Please keep one notation and check consistency in text, figures, and supplementary material. Such mixed notations affect the overall

consistency and quality of the document. Moreover, in figure 4, three experiments (pig 1, 2 and 3) are denoted. Even if this point can be clarified by authors quite easily, this supports my point for the major modification. The overall experimental procedure is unclear, and the reader has currently no guiding in the temporal relation between different results.

- Figures 4 and 5 are particularly dense and difficult to read. Authors may simplify the figure by showing only one animal result in the main figure and keep other for supplementary material.

- The statement on computational scaling on a quad core desktop (line 494) is fully critical. Even embarrassingly parallel computational problems do not follow fully linear relation between computational time and number of cores (Amdahl's law), and a clear limit due to memory exchanges and power dissipation is directly linked to the machine used. Please temper this statement.

- In equation (5) line 917, λ is not clearly defined. Please define and check that all notations in equation are explicitly defined.

- In figure 6, the spatial selectivity is not clearly elicited, however the method section on this metric is clear. Please add an explicit reference to this section.

- The online platform (reference 58) is not accessible, or at least I wasn't successful in accessing it with the given link. If it is proposed to the reader, you have to ensure one can access to the online resources, also as it is a citation, consider marking the code with a DOI to ensure a version can be referenced.

Reviewer #2 (Remarks to the Author):

The manuscript titled "Towards enhanced functionality of vagus neuroprostheses through in silico optimized stimulation" presents significant advancements in vagus nerve stimulation and computational neurostimulation modeling. It notably improves upon current methodologies by incorporating more anatomically realistic computational models, accelerating simulations, and computationally evaluating novel stimulation paradigms using data from experiments on pigs. The integration of histological fiber distribution within the vagus nerve is particularly commendable as it underscores the importance of histologically accurate models for VNS and tailored modeling. However, the manuscript overlooks the ramifications of this fiber distribution for electrode design, choice, and placement. Enriching the discussion on these implications would enhance an already robust manuscript.

The presented manuscript would further benefit from an expanded elaboration on computational methodologies and their limitations. A detailed explanation of the boundary conditions and their rationale is necessary, especially given their critical role in aligning computational models with electrophysiological realities. The limitations of the computational approach, including the impact of

longitudinal truncation on recruitment curves in light of potential end-node recruitment and the decision to use MRG models to reflect both afferent and efferent axons rather than specialized models that make a distinction between afferent and efferent axons, need to be thoroughly addressed. Moreover, the paper should discuss whether the substantial simulations conducted are essential for optimal results or if smaller subsets are generally sufficient. The assumption that action potential propagation correlates with the second spatial derivative of the extracellular potential and its generalizability across various nerve geometries also requires scrutiny.

The figures are of commendable quality and support the argumentation of the manuscript well. But, the manuscript requires proofreading to rectify grammatical and syntactical errors. Additionally, the citations are generally supportive but need better alignment with certain discussions within the text. Specific textual issues include:

-Line 34: missing “by” before B-efferents

-Line 33: The phrasing “Model unveils that all A-afferent fibers [...] influence laryngeal muscle [...]” is a very strong claim and should likely be adjusted to highlight that the model suggest the relationship between fiber recruitment and physiological outcome

-Line 65 missing “the” before “VN”

-Line 88 Discrepancies in fiber composition statistics in line 88, which should clarify the source species and its relevance.

-Line 91 missing “is” in front of “highly”

-Line 656 “telfon” instead of teflon

-Abbreviation “DB” in Line 671 hasn’t been introduced yet

-Line 935 A questionable selectivity index in line 935, lacking normalization or argumentation

-Throughout the text references to Fig 6e-g actually refer to Fig 6d-f (example: Line 1012)

-Additionally, the author contributions section requires revision to correct inconsistencies with listed acronyms and contributions. Issues such as the absence of co-author PB in the author list and the mismatch of initials for co-author NKS need rectification. Weiguo Song's contributions also need clarification.

Citations should be more appropriately aligned with the manuscript's content. For instance:

-The anatomical discussions of the vagus nerve in lines 47 – 49 are backed by publications focusing on bioelectric modulation rather than neuroanatomy, where neuroanatomical references would be more suitable.

-The third paragraph of the introduction (lines 68 – 71) would benefit from literature citations, especially where it asserts that current stimulation strategies fail to fully exploit potential fields possible with multipolar stimulation and that animal experiments are typically used to test new stimulation paradigms.

-In lines 96-98 the claim that “existing models still need hours to compute the estimated outcome, even with high-performance computing clusters [...]”, is something I am aware of in other computational neuroscience problems, but I am unable to find a reference on compute time as tested in high-performance computing clusters in the listed references. Additional citations may be beneficial.

Finally, I’d like to highlight that the online platform <https://neuroeng-hen.ethz.ch/online/> is out of service (tested twice over the course of one week)

Reviewer #2 (Remarks on code availability):

I was unable to find the code architecture on zenodo.

We thank the reviewers for their appreciation of our work and for very important and useful comments that helped us improve the quality of our manuscript. Our point-to-point replies (in blue), together with the modified text in the manuscript (in *blue italic*), to reviewers' comments (in black) are provided below. We also performed additional checks, corrected the typos and proof-read the manuscript by a native speaker.

Reviewer #1:

General remarks:

This contribution describes the development of realistic in silico models for vagus nerve stimulation. These models have realistic geometries and are optimized both to reproduce with high fidelity in vivo experimental observation and reduce the overall computational cost. Authors prove the viability of their model by proposing and validating a novel tri-polar stimulation strategy to control heart rate using VNS without reduced un-wanted motor or muscular activation. To do so, authors also include models for C-fibers which should be considered in VNS, adding to the novelty of the model. The overall work and results are impressive, of good quality and answer fundamental questions and challenge in the field of neuromodulation, and more specifically in field of vagus nerve stimulation. These elements justify in my opinion to accept this contribution for highly selective publication; however, authors should improve the overall completeness of the document on one major point and several minor points further detailed bellow.

We thank the reviewer for the appreciation of our study, and for the comments that helped us improving the manuscript. In the following revision, we are providing point-by-point responses to questions, together with emphasized changes we made in the manuscript.

Major modification:

Authors state on lines 538-539: "We have developed a novel method to personalize the models for a new animal undergoing experimental VNS testing". This sentence has a strong meaning, the word 'undergoing' meaning that in silico parameters exploration can be performed during experiments or surgery. Authors present some elements that clearly support this element, such as computational cost reduction (running the model is in the range of minutes instead of tens of hours). However, the overall experimental timeline is not clear and not sufficiently described and elicited. This setup is partially described in fig. 4.a, however one point is not clear and is in contradiction with the personalization (or the use of the word personalization by the authors): modeling part begins with a realistic model of cervical VN, however, the geometry is constructed from immunohistochemistry of nerve sample (described in method) that supposes the euthanasia of the subject. It is unclear if the realistic geometry is subject dependent, and in this case, models are not usable during undergoing experimental VNS testing, or if geometrical consideration are not subject specific and then the personalization has to be genuinely better described by authors. This should be clearly described in the text and supported by a figure describing the experimental path or timeline with the different in vivo/in silico phases, perhaps as an introduction to the method section.

We thank the reviewer for pointing this out. First, we developed two in-silico models of VNS based on histological images and immunohistochemistry of vagus nerve samples from two animals, labeled M1 and M2, ex-vivo (Figure 1). Then we performed in-vivo experiments on three new animals (S1, S2, and S3). We designed a protocol to adapt (personalize) these models to new experimental animals (S1-S3), in vivo, without requiring euthanasia of the subject, using only laryngeal EMG recordings easily obtained in vivo. This personalization is taking into account: i) that A α are among the largest fibers and therefore among the first to be electrically activated (*Raspopovic et al., 2012; Rattay, 1986; Veltink et al., 1988*) and ii) that activating A α efferent fibers triggers laryngeal muscle activation (*Blanz et al., 2023; Chang et al., 2020; Huffman et al., 2023; Jayaprakash et al., 2023; Nicolai et al., 2020; Qing et al., 2018; Yoo et al., 2013*). To account for the variation in the fiber thresholds caused by the inherent nerve structural variability across experimental subjects (e.g., a thicker epineurium), we scaled the model-predicted A α fibers response to match the recorded L-EMG responses. At the same time, we rotate the in-silico electrode position to reflect the in-vivo surgical placement. This results in the model predicted A α

responses aligned to the experimentally measured EMG for each active site. Finally, we validate the personalized models by comparing the experimentally recorded compound action potentials of A α and A β (CAP), per each active site, with in-silico estimated A α and A β fibers recruitment, finding high correlation (Fig. 4c). We made changes throughout text and figures to make the timeline clearer. The timeline and the experimental path are presented more in detail in the updated Figure 4 and its caption:

Figure 4. Model personalization and validation. (a) Experimental timeline and its steps: i. Construction of a realistic vagus nerve model from M1 and M2. ii. Experimental setup for the previously unseen subjects S1 - S3. iii. Personalization the models to the new experimental subjects, including the simple L-EMG recording setup, and customization of the model by rotation of active sites and scaling of thresholds. iv. Validation using CAP. (b) On the left, recruitment curves of M1 personalized to the three experimental subjects (S1, S2, S3), overlaid to the experimentally measured L-EMG, for three active sites. Results for all personalized models are presented in Supp. Fig. 6a. On the right: R^2 comparing personalized vs original linear regression between models and experimental subjects ($n = 48$, 6 pairs by 8 active sites). (c) i. In the first column, first row the normalized A α and A β CAP curves are reported per active site for S1. The corresponding recruitment curves of A α and A β estimated by the personalized M1 are reported in the second row. The second column represents the same data as heatmaps. ii. The normalized CAP from S1 and M1 model-predicted recruitment level for each active site and current level are correlated on a scatter plot. Pearson's correlation coefficient is reported. Results for all personalized models are presented in Supp. Fig. 6b. iii. The distribution of thresholds to recruit 10% of A α and A β fibers in all personalized models are compared to the experimentally measured thresholds to obtain a 10% of CAP value. iv. The ranking of active sites predicted by all personalized models is compared to the experiment by computing Spearman's correlation on thresholds, for recruitment levels between 10% and 70%. In the left box, it is reported the distribution of correlations for the personalized models ($n = 42$). In the right box, it is reported the distribution of correlation values obtained when shuffling the order of active sites, repeated 100 times ($n = 4200$).

We now labeled the three in-vivo experimental subjects as S1, S2, S3 through the text and figures to avoid confusion with the other two animals used to develop the realistic models (M1 and M2).

We updated Introduction:

We then designed a method to personalize the model to new subjects during in vivo experiments by matching the model-predicted fiber activation with the laryngeal electromyography (L-EMG), easily measurable during the surgery to accurately replicate the correct nerve-electrode interface, accounting for inherent inter-subject variability and surgical placement.

We updated Results:

To employ the developed models in-vivo on new experimental subjects and match physiological responses with the predicted fiber activation, we developed a methodology for personalizing the models using laryngeal electromyographic recordings (L-EMG) of the thyroarytenoid muscle (Fig. 4a), easily measurable during the surgery. This method is based on the fact that these large fibers are the first to be recruited by electrical stimulation (Raspovic et al., 2012; Rattay, 1986; Veltink et al., 1988) and the assumption that activation of A α efferent fibers induces the laryngeal muscle response (Blanz et al., 2023; Chang et al., 2020; Huffman et al., 2023; Jayaprakash et al., 2023; Nicolai et al., 2020; Qing et al., 2018; Yoo et al., 2013). We rotate the electrode with the modeled active sites (thanks to the axial symmetry of the cuff electrode) to mimic possible variations during surgical placement, and scale model-estimated thresholds to account for individual variability at the nerve-electrode interface. We applied this method to personalize both M1 and M2 to three new experimental subjects (S1, S2, and S3).

We reordered the sections in Methods to align them with the actual timeline, and expanded some paragraphs:

We developed a method to adapt these highly detailed computational models to new experimental subjects for which no histology can be obtained (in vivo). [...] We computed the personalization steps for each model (M1 and M2) and experimental subject (S1, S2, and S3) for a total of 6 personalized pairs of models-experimental subject.

Finally, we updated Fig. 1 a to improve the clarity of the pipeline implemented to develop the realistic models and the required data:

Figure 1. Pipeline for creating and exploiting histologically and morphologically realistic model of cervical vagus nerve stimulation (VaStim). (a) The 3D nerve reconstruction is based on histological images of VN cross-sections, considering fascicles' curvatures, branching and merging along the nerve. Immunohistochemistry is applied to determine the precise location, type (A α , A β , A γ , A δ , B, C) and modality (afferent, efferent) of different fiber types and the VN is populated with fibers, following fascicle structure along the nerve. We modeled helical nerve cuff that is used in the VNS animal experiments we performed.

Minor modifications:

- Authors refer in the text to two animal models that are named M1 and M2 (line 136), however they are renamed A1924 and A6050 (Fig. 3 for instance but not only). Please keep one notation and check consistency in text, figures, and supplementary material. Such mixed notations affect the overall consistency and quality of the document. Moreover, in figure 4, three experiments (pig 1, 2 and 3) are denoted. Even if this point can be clarified by authors quite easily, this supports my point for the major modification. The overall experimental procedure is unclear, and the reader has currently no guiding in the temporal relation between different results.

We thank the reviewer for noticing this inconsistency in labels. We updated main and supplementary figures to unify the nomenclature. The issue about the three experimental subjects is explained in the detailed answer to the previous question and the related manuscript variations.

- Figures 4 and 5 are particularly dense and difficult to read. Authors may simplify the figure by showing only one animal result in the main figure and keep other for supplementary material.

We appreciate this suggestion to improve the clarity of figures. We implemented it producing new figures 4 (reported above) and 5 and expanding corresponding supplementary figures 6 and 7.

Figure 5. Explanation of experimental heart rate variation through in-silico models and improvement of selectivity through tripolar stimulation. (a) Pipeline to optimize HR changes while minimizing L-EMG side effects. Fascicles of the personalized models that are responsible for experimental HR changes are extracted. Then different stimulation paradigms can be tested on the personalized models, and HR vs L-EMG recruitments are computed. **(b)** Finding heart-specific fascicles M1 personalized to S1. Level of HR change (left, bars) caused by fiber activation in specific fascicles. Fascicles are spatially presented and color-coded based on these values (right). Results for all

personalized models are reported in Supp. Fig. 7a. (c) Recruitment curves of M1 personalized to S1, overlaid to the experimentally measured Δ HR, for active sites with recruitment higher than 0 in the experimental range. Results for all personalized models are reported in Supp. Fig. 7b. (d) Top part: example of monopolar (left) vs tripolar (right) stimulation. For each stimulation paradigm the normalized potential and the Δ HR B_{Eff} and L-EMG A α activations are shown. Bottom part: the off-target (A α) recruitment and target (B_{Eff}) recruitment between monopolar and tripolar case. (e) Experimental recruitment curves of Δ HR and L-EMG. (f) Comparison of monopolar vs tripolar off-target (A α) recruitment at 25% recruitment of B_{Eff} (model-estimated fiber activation at experimentally obtained threshold stimulation amplitude, causing clinically relevant Δ HR) for all pairs model-experimental subject ($p = 0.031$, $n = 6$, $z = 2.2$). (g) Comparison of monopolar vs tripolar charge to recruit 25% of B_{Eff} (targeted Δ HR) for all pairs model-experimental subject ($p = 0.031$, $n = 6$, $z = -2.2$).

Supplementary Figure 6. Additional results on model personalization and validation. (a) Recruitment curves of M1 and M2 personalized to the three experimental subjects, overlaid to the experimentally measured L-EMG, for all active sites. (b) i. One row per subject, in the first and second column the normalized fast CAP curves are reported per active site are reported. In the third and fourth column, the corresponding recruitment curves of A α and A β estimated by the personalized M1. In the fifth and sixth column, the same for M2. ii. The normalized CAP and M2 model-predicted recruitment level for each active site and current

level are correlated on a scatter plot. Pearson's correlation coefficient is reported. **iii.** The distribution of thresholds to recruit 10% to 70% of A α and A β fibers in the personalized models are compared to the experimentally measured thresholds to obtain a CAP value of 10% to 70% (n = 48 per boxplot). **iv.** The ranking of active sites predicted by the personalized models is compared to the experiment by computing Spearman's correlation on thresholds, for recruitment levels between 10% and 70% (n = 42 per boxplot).

Supplementary Figure 7. Additional results regarding prediction of heart rate variation through personalized models (a) Finding heart-specific fascicles for both models personalized to the three experimental subjects. Level of HR change (left, bars) caused by fiber activation in specific fascicles. Fascicles are spatially presented and color-coded based on these values (right). **(b)** Recruitment curves of the models personalized to the three experimental subjects, overlaid to the experimentally measured Δ HR, for all active sites.

- The statement on computational scaling on a quad core desktop (line 494) is fully critical. Even embarrassingly parallel computational problems do not follow fully linear relation between computational time and number of cores (Amdahl's law), and a clear limit due to memory exchanges and power dissipation is directly linked to the machine used. Please temper this statement.

We apologize for this approximation and acknowledge the reviewer concerns. We tempered and extended the text as follows:

The time necessary to perform these simulations is reduced by multiprocessing since the computation of single fiber responses is an embarrassingly parallel problem. However, since the computational benefit is less than linear with the number of available processing threads (Amdahl, 1967), the simulations would also be performed in a reasonable time span on a standard quad-core desktop computer.

Since computational efficiency is less than linearly proportional to the number of threads, as reviewer correctly observed, other resources equal, the computation time on a quad core computer would be less than 12 times larger than what we observed with a 48-core computer (less than 80 minutes), which motivates our statement.

- In equation (5) line 917, lambda is not clearly defined. Please define and check that all notations in equation are explicitly defined.

We thank the reviewer for pointing this up. We have defined the lambda in the mentioned equation:

where $\delta_{\lambda,mod}$ corresponds to the relative deviation, λ_{mod} to the recruitment threshold obtained with the optimized method, and λ_{orig} to the recruitment threshold obtained with original method.

We also added other missing definitions across all equations (in blue in the revised manuscript).

- In figure 6, the spatial selectivity is not clearly elicited, however the method section on this metric is clear. Please add an explicit reference to this section.

We have expanded the Figure 6 caption with the short explanation of the proposed metric and referenced Methods section for more details:

Representation of the two extreme cases of the spatial selectivity index $\eta \in [-1,1]$. The index is calculated for a targeted fascicle (contoured in blue). Red fascicles indicate recruitment of all contained fibers. $\eta = 1$ corresponds to the activation of all fibers in the targeted fascicle without the recruitment of any other fiber, $\eta = -1$ to the activation of all fibers in all fascicles except the targeted one (see Eqn. 8).

Moreover, we added a citation to the related Method section to a previous publication describing this index (Raspovic et al., 2017).

- The online platform (reference 58) is not accessible, or at least I wasn't successful in accessing it with the given link. If it is proposed to the reader, you have to ensure one can access to the online resources, also as it is a citation, consider marking the code with a DOI to ensure a version can be referenced.

We thank the reviewer for reporting this technical issue which has been solved. We share the concern about perpetual availability; therefore we included the complete source code of the published online platform, including the data required for visualization, in the revised Zenodo publication at <https://zenodo.org/records/11219384>. Additionally, we also provided a live distribution of the online visualization platform at <https://neuroeng-hen.ethz.ch/online/> to allow the visualization of the models without compilation. This way, no reference to non-permanent resources is made throughout manuscript.

Reviewer #2:

General remarks:

The manuscript titled "Towards enhanced functionality of vagus neuroprostheses through in silico optimized stimulation" presents significant advancements in vagus nerve stimulation and computational neurostimulation modeling. It notably improves upon current methodologies by incorporating more anatomically realistic computational models, accelerating simulations, and computationally evaluating novel stimulation paradigms using data from experiments on pigs. The integration of histological fiber distribution within the vagus nerve is particularly commendable as it underscores the importance of histologically accurate models for VNS and tailored modeling. However, the manuscript overlooks the ramifications of this fiber distribution for electrode design, choice, and placement. Enriching the discussion on these implications would enhance an already robust manuscript.

We thank the reviewer for highlighting the advances in our work and for providing useful suggestions.

We appreciate the suggestion to enrich the discussion with a discussion on the implications of the fiber distribution. We therefore added a paragraph discussing the implications of accurate fiber distribution and its ramifications:

[...] this evidence supports the choice of eight active sites for a cuff electrode, suggesting that these sites adequately cover the circumference of the nerve, regardless of their orientation (Jayaprakash et al., 2023). On the other hand, the fact that only very poor selectivity can be achieved for non-superficial fascicles, combined with the observed functional clustering, suggests that intraneural electrodes may be required to modulate functions which cluster deeper within the nerve, and indeed has been shown promise for VNS (Agnesi et al., 2023). Moreover, the use of intrafascicular electrodes can improve the selectivity for smaller diameter axons (Ciotti et al., 2023), especially if they cluster within fascicles, possibly reducing off-target effects such as laryngeal muscle contractions due to A_{eff} recruitment. The strong functional implications of spatial clustering indicate that the accurate fiber distribution is crucial to consider when utilizing models to design electrodes and stimulation paradigms, and generally reinforces the importance of spatial selectivity for obtaining therapeutic effects minimizing undesirable side effects.

And:

Moreover, VaStim can be used as a platform to design and optimize electrode designs, as explored in previous studies (Aristovich et al., 2021; Ciotti et al., 2023), enriched by the accurate representation of clustering of fiber types and their functionalization through physiological experiments.

The presented manuscript would further benefit from an expanded elaboration on computational methodologies and their limitations. A detailed explanation of the boundary conditions and their rationale is necessary, especially given their critical role in aligning computational models with electrophysiological realities.

We thank the reviewer for these relevant comments and suggestions. We reported all boundary conditions in Methods, which we slightly expanded for completeness:

In order to solve the electrical fields that the active sites induce during stimulation, the 3D model was imported into COMSOL Multiphysics. Here a homogeneous saline cylinder with radius of 35 mm was added to emulate the interoperative environment, with the diameter chosen by a convergence study so that the ground-at-infinity condition was sufficiently approximated (Cimolato et al., 2023; Ciotti et al., 2021; Raspopovic et al., 2011, 2017). The perineurium was modeled by adding a thin layer contact impedance to each fascicle boundary, with the surface thickness being set to 3% of the average area over all four cross sections of the corresponding fascicle. This approach leaned on the common assumption that the endoneurium thickness equals about 3% of the fascicle area (Ciotti et al., 2021, 2023; Grinberg et al., 2008; Raspopovic et al., 2017; Zelechowski et al., 2020). Epineurium, perineurium, electrode substrate and saline were assumed to be isotropic, whereas the endoneurium was modeled with anisotropic conductivity, due to the presence of fibers within. The precise conductivity values are based on previous modeling studies (Raspopovic et al., 2017), and listed in Supp. Table 2. To ensure that the anisotropy is correctly oriented with respect to the fascicular cross sections, and not the global coordinate system, we used COMSOL's Curvilinear Coordinates feature. In particular, the anisotropic vector field was aligned to the streamlines determined by a diffusion study, in which the fascicle hulls were defined as walls and their flat ends as inlets and outlets. Both nerve models were meshed resulting in about 15 to 20 million

tetrahedral elements. For low frequency stimulation, it is adequate to exploit the quasi-static approximation of Maxwell's equations in order to calculate the electric fields V_{ext} (Bossetti et al., 2007) (Eqn. 1), where σ is the conductivity:

$$\nabla \cdot \sigma \nabla V_{\text{ext}} = 0 \quad (1)$$

As this equation is linear, it is possible to determine the electric fields resulting from an arbitrary multipolar stimulation by scaling and superimposing electric fields which were independently determined for all active sites under consideration of unit charge injection (Raspopovic et al., 2017) *from surface current sources corresponding to each active site.*

We integrated additional discussion about the computational methodology including boundary conditions in Limitations regarding the use of a homogeneous surrounding medium:

As physical models are always partial representations of reality, a compromise between accuracy and simplicity must be drawn, as too complex models can be unjustifiably costly to develop or simulate, while too simple ones can lead to misleading conclusions. Simplifications include the representation of the surrounding medium as homogeneous saline environment as approximation of the operating conditions, as typically done (Cimolato et al., 2023; Ciotti et al., 2021, 2023; Raspopovic et al., 2011, 2017; Romeni et al., 2020; Schiefer et al., 2008; Zelechowski et al., 2020). Representing precisely the surrounding tissues which would have made recruitment estimations more accurate, but it would have likely not affected the essential conclusions (Bucksot et al., 2019). Moreover, the personalization step we devised should be able to at least partly account for this type of discrepancies.

The limitations of the computational approach, including the impact of longitudinal truncation on recruitment curves in light of potential end-node recruitment and the decision to use MRG models to reflect both afferent and efferent axons rather than specialized models that make a distinction between afferent and efferent axons, need to be thoroughly addressed.

Regarding longitudinal truncation, we agree that the choice of improper truncation length can lead to errors. Indeed, we performed a convergence study (see Fig. 2c-d), to ensure that the modeled length of the fibers were sufficient to avoid spurious effects at the extremities (more precisely, it ensures that recruitment thresholds errors are below 0.5% for at least 99.5% of fibers). However, we added a sentence in limitations explaining the possible need to repeat the convergence studies in case of much different nerve and electrode configurations:

Also, in case of greatly different configurations (e.g., with intrafascicular electrodes), it may be beneficial to repeat the convergence studies for dynamic discretization (Fig. 2a-b) and longitudinal truncation (Fig. 2c-d).

We agree that using different models for afferent and efferent fibers could potentially even improve the accuracy of the model even more. Indeed, during preliminary tests we also implemented modality-specific models such as Gaines et al. 2018, which however resulted in very moderate differences, while being too computationally costly with respect to MRG. However, we included a sentence in Limitations regarding it:

As physical models are always partial representations of reality, a compromise between accuracy and simplicity must be drawn, as too complex models can be unjustifiably costly to develop or simulate, while too simple ones can lead to misleading conclusions. [...] Even higher prediction accuracy may also be potentially obtained by using modality-specific models for afferent and efferent fibers (Eiber et al., 2021).

Moreover, the paper should discuss whether the substantial simulations conducted are essential for optimal results or if smaller subsets are generally sufficient. The assumption that action potential propagation correlates with the second spatial derivative of the extracellular potential and its generalizability across various nerve geometries also requires scrutiny.

Regarding the importance of these substantial simulations, we agree that using a small subsample of fibers is often sufficient, depending on the objective. However, we first intended to show that full-scale simulations are feasible, useful for example in case the activity of each single axon is of interest, which is

not absurd considering for example that humans can perceive sensations even from single afferents (Sanchez Panchuelo et al., 2016). Only then we performed a subsampling study to reduce the number of simulated fibers while limiting the mean error in recruitment curves under 2% (Supplementary Figure 1). It resulted in a subsample of about 10'000 fibers, with respect to the few hundred thousand of the complete populations. We added a sentence in Discussion regarding the population size:

We demonstrated that by using VaStim it is feasible in short time spans to execute large-scale studies of vagus nerve stimulation, containing more than 10 thousand fibers of all types *for accurate estimation of recruitment, and up to the whole fiber population of the vagus nerve when the activity of each single axon is of interest.*

Regarding the use of the second derivative of extracellular potential, while we agree with reviewer that it is not a perfect predictor of subthreshold membrane depolarization,, we just used it as a tool to more optimally discretize unmyelinated fibers into sections (Fig. 2a), as opposed to the usual method of discretizing them in fixed-length sections. So, it was only as auxiliary for the discretization of detailed simulations, and after that we do not refer anymore to the AF. We performed a convergence study to choose the parameters of the discretization so that the error on threshold is limited ($0.4 \pm 1.6\%$ deviation with respect to fixed-length discretization, Fig. 2b). However, we agree that in case of much different configurations, where the shape of the extracellular potential may be much different from our case, it may be beneficial to repeat the convergence studies, so we included a paragraph in Limitations (reported earlier in this section).

The figures are of commendable quality and support the argumentation of the manuscript well. But, the manuscript requires proofreading to rectify grammatical and syntactical errors. Additionally, the citations are generally supportive but need better alignment with certain discussions within the text.

We performed proofreading and update citations throughout the text.

Specific textual issues include:

-Line 34: missing "by" before B-efferents

We have corrected the mistake.

-Line 33: The phrasing "Model unveils that all A-afferent fibers [...] influence laryngeal muscle [...]" is a very strong claim and should likely be adjusted to highlight that the model suggest the relationship between fiber recruitment and physiological outcome

We agreed and replaced unveils with suggests.

-Line 65 missing "the" before "VN"

We thank the reviewer for noticing this mistake, we have corrected it.

-Line 88 Discrepancies in fiber composition statistics in line 88, which should clarify the source species and its relevance.

We thank the reviewer for this notice, we corrected the reference and included the range of C-fiber percentage in the VN.

-Line 91 missing "is" in front of "highly"

-Line 656 "telfon" instead of teflon

We corrected these mistakes.

-Abbreviation "DB" in Line 671 hasn't been introduced yet

The "DC" abbreviation in line 671 refers to the "direct current" component of the signal. We slightly rephrased for clarity:

After removal of the direct current (DC) component.

-Line 935 A questionable selectivity index in line 935, lacking normalization or argumentation

We added reasoning to the choice of this index and an explanation of its range:

Such metric is commonly used (Kent & Grill, 2013; Schiefer et al., 2008) and is easily interpretable. As other spatial selectivity indexes (as in Eqn. 8), it ranges from -1 (recruitment of all non-target fibers, no recruitment of target fibers), to 1 (recruitment of all target fibers, no recruitment of non-target fibers).

-Throughout the text references to Fig 6e-g actually refer to Fig 6d-f (example: Line 1012)

We thank the reviewer for this remark, we corrected it throughout the manuscript.

- Additionally, the author contributions section requires revision to correct inconsistencies with listed acronyms and contributions. Issues such as the absence of co-author PB in the author list and the mismatch of initials for co-author NKS need rectification. Weiguo Song's contributions also need clarification.

We apologize for this mistake. NKS corresponds to the author Natalija Katic Secerovic, we have corrected the author's surname in the list of authors. We also added contributions of the authors Weiguo Song and Andrea Cimolato and moved the incorrect reference to PB to Acknowledgments.

Citations should be more appropriately aligned with the manuscript's content. For instance:
-The anatomical discussions of the vagus nerve in lines 47 – 49 are backed by publications focusing on bioelectric modulation rather than neuroanatomy, where neuroanatomical references would be more suitable.

We thank the reviewer for identifying this misalignment. We reorganized these references and placed two more appropriate neuroanatomical references (Rea 2014, and Yuan and Silberstein 2016).

-The third paragraph of the introduction (lines 68 – 71) would benefit from literature citations, especially where it asserts that current stimulation strategies fail to fully exploit potential fields possible with multipolar stimulation and that animal experiments are typically used to test new stimulation paradigms.

We reworded the paragraph and added two references to clarify this point:

Considering the sheer multidimensionality of the stimulation parameters space, exploring complex VNS paradigms in-vivo is nearly impossible and thus constrained to explore just a sub-optimal subset of possible policies, such as only bipolar (Shulgach et al., 2021). [...] Previous studies have leveraged computational models to design more spatially selective electrodes and stimulation policies. However, due to the lack of accurate information regarding fiber distributions and function, animal experiments were necessary to assess the physiological effects of novel stimulation paradigms (Aristovich et al., 2021; Dali et al., 2018).

-In lines 96-98 the claim that "existing models still need hours to compute the estimated outcome, even with high-performance computing clusters [...]", is something I am aware of in other computational neuroscience problems, but I am unable to find a reference on compute time as tested in high-performance computing clusters in the listed references. Additional citations may be beneficial.

There is not a large literature body regarding the use of high performance clusters in this kind of applications, and the sentence was based on our previous experience and data reported in the beginning of Results section regarding computational time before optimizations. Anyways, we added a citation to Pelot et al. 2015 reporting that 3 hours were needed to simulate unmyelinated axons on a HPC with an MRG-like model.

Finally, I'd like to highlight that the online platform <https://neuroeng-hen.ethz.ch/online/> is out of service (tested twice over the course of one week)

We thank the reviewer for reporting this technical issue which has been solved. We would like to highlight that we also included a complete build of the platform in the revised Zenodo submission for permanent archiving (<https://zenodo.org/records/11219384>).

Remarks on code availability:

I was unable to find the code architecture on zenodo.

The upload on Zenodo was still restricted, we apologize for this mistake. It is now public at <https://zenodo.org/records/11219384>.

REVIEWERS' COMMENTS

Reviewer #1 (Remarks to the Author):

I thank the authors for the careful review and consideration of suggested modifications.

The timeline of experiments and the new notation of subjects M1 and M2 for geometry extraction and S1, S2, S3 make the full publication more readable. The proposed work is interesting and of high scientific quality and consistency.

After a careful proofread, I have, as reviewer, no scientific objection for publication. The topic of electrical stimulation of Vagus nerve is challenging and the contribution is significant.

Reviewer #2 (Remarks to the Author):

The revised manuscript is very well done and has significantly improved. The authors have addressed the majority of my comments thoroughly, and the manuscript is much clearer and more robust. I commend the authors for their diligent revisions and attention to detail. However, the manuscript may still benefit from very few minor adjustments:

My comment that "The assumption that action potential propagation correlates with the second spatial derivative of the extracellular potential [...] also requires scrutiny" can be further addressed. While I appreciate the additions to the limitations section, the revised manuscript may be further improved by changing the phrasing in lines 198-200 of the revised manuscript, that generically states that it is a fact that action potential generation is proportional to the second spatial derivative of the extracellular potential along the fiber. While it is true that the activation function, which is proportional to the second spatial derivative of the extracellular field potential, serves as a predictor for neural activation, it does not account for the shape of stimulation pulses, transient transmembrane potential evolution, interactions between neighboring fiber sections, or activation at fiber terminals. A simple rephrasing in lines 198-200 to weaken the factual nature of the statement or remove it altogether would strengthen the manuscript.

Additionally, in Figure 5, panel d, there is likely a spelling mistake in the word "undersired," which the authors likely meant to write as "undesired".

These minor adjustments would help in making the already robust manuscript more precise and polished.

Reviewer #2 (Remarks on code availability):

The README simply states to run a .bat file which is windows script, so I did not test the code on my MAC.